# Quantifying the changes of soil surface microroughness due to rainfall-induced erosion on a smooth surface

Benjamin K.B. Abban[1], A. N. (Thanos) Papanicolaou[1,5], Christos P. Giannopoulos[1], Dimitrios C. Dermisis[2], Kenneth M. Wacha[3], Christopher G. Wilson[1], Mohamed Elhakeem[4]

[1]Hydraulics and Sedimentation Lab, Department of Civil & Environmental Engineering, University of Tennessee – Knoxville, Knoxville, TN 37996 USA
[2]College of Engineering, Department of Chemical, Civil & Mechanical Engineering, McNeese State University, Lake Charles, LA 70605 USA
[3]USDA-ARS – National Laboratory for Agriculture and the Environment, Ames, IA 50011 USA
[4]Abu Dhabi University, Abu Dhabi, P.O. Box 59911, United Arab Emirates
[5]Tennessee Water Resources Center, Knoxville, TN 37996 USA

*Correspondence to*: Prof. Athanasios (Thanos) N. Papanicolaou (tpapanic@utk.edu)

**Abstract.** This study examines the rainfall induced change in soil microroughness of a bare smooth soil surface in an agricultural field. Smooth soil surfaces with microroughness on the order of ~2-5 mm are common in agricultural landscapes subject to long, undisturbed exposure to rainfall impact and runoff and freeze-thaw cycles. There are quantitative indications in the literature that under such conditions, roughness may increase subject to rainfall action. The focus is on the quantification of soil surface roughness due to rainfall for initial microroughness length scales of 2 mm or less, which represent generic extreme conditions. These conditions have not been extensively examined in the literature as most studies have focused on disturbed initial surface conditions with roughness on the order of ~5-50 mm. Three rainfall intensities of 30 mm/h, 60 mm/h and 75 mm/h were applied to a smoothened bed surface in a field plot via a rainfall simulator. These intensities represent the range from typical to extreme rainfall intensity conditions that appear in the region of study. Soil surface elevations were obtained via a surface-profile laser scanner. Several indices were utilized to quantify soil surface microroughness, namely the Random Roughness (RR) index, the crossover length, the variance scale from the Markov-Gaussian model, and the limiting difference. Findings show a consistent increase in roughness under the action of rainfall, with higher rainfall intensities resulting in higher relative roughness increase. This contradicts the commonly adopted notion in existing literature that a monotonic decay of soil surface roughness with rainfall is expected regardless of initial surface roughness conditions. The study results highlight the need for a better understanding of the phenomenon of microroughness evolution on a bare surface under rainfall action and its potential implications on hydrologic response.

## 1 Introduction

Soil surface roughness influences many hydrologic processes such as flow partitioning between runoff and infiltration, flow unsteadiness, as well as soil mobilization and redeposition at scales ranging from a few millimeters to hillslope level (e.g. Huang and Bradford, 1990; Magunda et al., 1997; Zhang et al., 2014).

There are three distinct classes of microtopography surface roughness (Fig. 1a) for agricultural landscapes, each one of them depicting a representative length scale (Römkens and Wang, 1986; Potter, 1990). Following Oades and Waters (1991), the first class includes microrelief variations from individual soil grains to aggregates in the order of 0.053-2.0 mm. The second class consists of variations due to soil clods ranging between 2-100 mm. The third class of soil surface roughness is systematic elevation differences due to tillage, referred to as oriented roughness (OR), ranging between 100-300 mm.

From the outlined above, the first two classes are the so called random roughness (RR), and constitute the main focus of the present research. Contrary to OR, which changes seasonally and during crop rotations, RR changes on an event base (Abaci and Papanicolaou, 2009). RR reflects the effects of rainfall action on the soil surface and inherently varies in space and time. As a result, RR affects key hydrologic processes at the soil scape and ultimately at the hillslope scale (e.g., overland flow, infiltration), by affecting the depression storage and the associated runoff and erosion processes (Gómez and Nearing, 2005;

Chi et al., 2012). According to Paz-Ferreiro et al. (2008), the RR index, which was first proposed by Allmaras et al. (1966), is the most widely used statistical microrelief index for the evaluation of soil surface roughness. The RR index was initially calculated per Allmaras et al. (1966) as the standard deviation of the log-transformed residual point elevation data. In this study, it is calculated according to Currence and Lovely (1970) as the standard deviation of bed surface elevation data around the mean elevation, after correction for slope using the best fit plane and removal of tillage effects in the individual

height readings:

$$RR = \sqrt{\frac{\sum_{i=1}^{n}(Z_i - \bar{Z})^2}{n}}, \tag{1}$$

where $Z_i$ and $\bar{Z}$ are individual elevation height readings and their mean, respectively, and $n$ is the total number of readings. Characterization of RR remains a challenge. Most of the available studies are limited to soil surfaces where the length scale exceeds the upper microrelief length scale of 2 mm corresponding to the first class. These studies usually include bed surface

conditions with initial length scales of 5-50 mm (e.g., Zobeck and Onstad, 1987; Gilley and Finkner, 1991). In these studies, a monotonic decay of roughness due to precipitation action is predicted, since rainfall impact and runoff "smoothen" the rough edges of soil grains, aggregates and clods, especially in the absence of cover (Potter, 1990; Bertuzzi et al., 1990; Vázquez et al., 2008; Vermang et al., 2013). Few existing studies, to the best of our knowledge, have explicitly examined the interaction of raindrop impact with bare soil surfaces for initial microroughness scales on the order of ~2-5 mm. Surfaces

with microroughness on the order of ~2-5 mm are common in agricultural landscapes where the soil is "smoothened" due to long, undisturbed exposure to rainfall impact, runoff, and freeze-thaw cycles (Burwell et al., 1963; Allmaras et al., 1966; Burwell et al., 1969; Cogo, 1981; Currence and Lovely, 1970; Steichen, 1984; Unger, 1984; Zobeck and Onstad, 1987; Abaci and Papanicolaou, 2009). Within these landscapes, soil surface conditions are usually bare in the period of the crop

rotation between post-harvest and before plant growth is established, which approximately corresponds to 30-75% of the cyclic crop rotation period.

There are some quantitative indications that under bare smooth surface conditions, soil surface roughness may actually increase under the action of rainfall. Specifically, the study by Huang and Bradford (1992) calculated the semivariance with respect to length scale before and after rainfall, and a slight increase in roughness with rainfall was denoted using the Markov-Gaussian model for a surface with low initial roughness. Rosa et al. (2012) introduced an index (called Roughness Index) estimated from the semivariogram to describe roughness, and an increase of the index with rainfall was observed under specific conditions, and attributed to the fragmentation of aggregates and clods to smaller aggregates. Zheng et al. (2014) reported an increase in values of the RR after the application of rainfall on smooth soil surfaces. Finally, Vázquez et al. (2008) examined the evolution of the surface of three different soils during successive events. They reported that for two out of three soils, roughness increased for the first event, however decreased for the following events; the third soil showed scarce trend to either increasing or decreasing roughness due to successive rainfall events. Nevertheless, none of the above studies explicitly stated or acknowledged the increasing trend of roughness and its potential linkage to smooth bare soil surface conditions. We herein examine changes in RR under rainfall impact for initial microroughness of the order of 2 mm. The main goal is to examine the postulate observed in the literature (e.g., Zheng et al., 2014) that rainfall action on surfaces with initial microrelief on the order of 2-5 mm could lead to RR increase. An increase in RR under these scales would further imply that rainfall action cannot completely eliminate the roughness of a surface, so roughness residuals would always exist at the locations where raindrop detachment is prevalent. Hereafter, for shortness, tests with initial RR on the order of 2-5 mm will be referred to as "smooth", whereas tests with initial RR greater than 5 mm will be referred to as "disturbed".

The key specific objectives of this study are (i) to quantify the soil surface microroughness of smooth bare soil surfaces before and after the effect of rainfall, and (ii) calculate the relative change in roughness for different intensities. To meet the two specific objectives we employ four commonly used indices, the RR index, the crossover length, the variance scale from the Markov-Gaussian model, and the limiting difference. The last three indices are alternate methods and used here to supplement the RR index analysis for relative change in roughness.

## 2 Materials and Methods

### 2.1 Experimental Conditions

This study was conducted on an experimental plot (Fig. 1b) of the U.S. National Science Foundation Intensively Managed Landscapes Critical Zone Observatory in the headwaters of Clear Creek, IA (41.74º N, -91.94º W and an elevation of 250 m above mean sea level). The soil series at the plot where the experiments were conducted is Tama (fine-silty, mixed, superactive, mesic Cumulic Endoaquoll) (http://criticalzone.org/iml/infrastructure/field-areas-iml/). It consists of 5% sand, 26% clay, 68% silt, and an organic matter content of 4.4%. The aggregate size distribution of the soil consists of 19% of the

soil size fraction less than 250 μm, 48% between 250 μm and 2 mm, and 33% greater than 2 mm. These soils contain both smectite and illite, with high cation exchange capacity between 15 and 30 Meq/100 g. The experimental plot was uniform in terms of downslope curvature, its gradient was 9% and the plot size was approximately 7 m long by 1.2 m wide.

The soil surface was prepared before each experiment by tamping using a plywood board to create a smoothened surface. Rainfall was applied to the plot using Norton Ladder Multiple Intensity Rainfall Simulators designed by the USDA-ARS National Soil Erosion Research Laboratory, IN. Figure 2 shows the setup for all the experimental runs considered in the present study. For each test, three rainfall simulators were mounted in series over the experimental plot (Fig. 2a) and approximately 2.5 m atop the plot surface (Fig. 2b) in order to ensure that raindrop terminal velocity was reached. Water was continuously pumped from a water tank under controlled pressure, and uniform rainfall was applied through oscillating VeeJet nozzles which provided spherical drops with median diameters between 2.25-2.75 mm and a terminal velocity between 6.8-7.7 m/s depending on the rainfall intensity. The distribution of raindrop sizes generated by the rainfall simulators was calibrated using a disdrometer and followed a Marshall-Palmer distribution (Elhakeem and Papanicolaou, 2009), which is a widely accepted distribution for natural raindrop sizes in the U.S. Midwest where the study was performed (Marshall and Palmer, 1948). The calibration of the raindrop sizes was achieved by adjusting the pressure and swing frequency of the VeeJet nozzles. This level of attention was taken to minimize any potential biases compared to natural rainfall with respect to raindrop size distribution, and, thus, render the rainfall simulation experiments scalable to other regions experiencing the same type of soil, bare surface, roughness conditions, and natural rainfall characteristics.

Surface elevations were obtained prior to and after the completion of the experiments via an instantaneous digital surface-profile laser scanner (Darboux and Huang, 2003), developed by the USDA-ARS National Soil Erosion Research Laboratory, IN (Fig. 3a). Laser scanner measurements before the runs confirmed that the overall microrelief was less than 2 mm. Horizontal and vertical accuracies of the laser are 0.5 mm. Thus, microroughness features less than 0.5 mm may not have been captured in the analysis. Points were measured every 1 mm. The system consists of two laser diodes mounted 40 cm apart to project a laser plane over the targeted surface. The beam is captured by an 8-bit, high-resolution progressive scan charge-couple device camera with 1030 rows x 1300 columns and a 9 mm lens. The camera and lasers are mounted on a 5 m long carriage assembly and their movement on the carriage is controlled by software that regulates the travel distance based on a user-specified distance (Fig. 3a). Information captured by the camera is recorded with an attached computer. The information from each scan is converted into a set of (x,y,z) coordinates using a calibration file and the software developed from the USDA-ARS National Soil Erosion Research Laboratory for data transformation as explained by Darboux and Huang (2003). The set of (x,y,z) coordinates obtained for each experiment are imported into ArcGIS 10.3.1 in order to create the corresponding Digital Elevation Models (DEMs) through inverse distance weighting interpolation and thereby visualize or analyze the surfaces (Fig. 3b). The resulting DEMs have a horizontal resolution of 1 mm and an accuracy of 0.5 mm in the vertical.

Three tests of varying rainfall intensity were conducted on the experimental plot. Rainfall intensities were respectively 30, 60 and 75 mm/h for experiments 1, 2 and 3. These simulated intensities represent typical storms observed in the region of

South Amana where the plot is located (Huff and Angel, 1992). Three replicates of each rainfall intensity case were performed until steady state conditions, and repeatability was confirmed by evaluation of changes in RR at specific cross-sections in the rainsplash dominated zone. It was found that on an average, the relative error of the RR ratios between replicates did not exceed 7%. The volumetric water content was recorded via six 5TE soil moisture sensors manufactured by Decagon Devices, Inc. and placed along the plot to a depth of 10 mm. The initial volumetric water content was found to be similar for each experiment and approximately equal to 35% at the whole plot, where the field capacity of the specific soil is 38%. Each experiment was run for nearly 5 hours, sufficiently long to reach steady state conditions, as confirmed by weir readings and discrete samples taken at the outlet of the plot. The infiltration rate was estimated during all rainfall simulation runs by subtracting the measured runoff rates from the constant rainfall rates. This approach has been commonly used in plot experiments and provides a good estimate of the spatially averaged infiltration rates (e.g., Mohamoud et al., 1990; Wainwright et al., 2000). Averaged saturated hydraulic conductivity values ranged from 3.20 – 4.56 mm/h, which are in agreement with the averaged saturated hydraulic conductivity value of 4.3 mm/h measured by Papanicolaou et al. (2015a) using semi-automated double ring infiltrometers at the field where the study was performed.

The initial microroughness length scale in Experiment 1 (1.17 mm) was greater than that of Experiment 2 (0.42 mm) and Experiment 3 (0.32 mm) – see Table 1. This is attributed to the different timing of the experiment runs with respect to tillage. Experiment 1 was performed in early August, soon after harvest, so the soil surface had recently been disturbed. However, for Experiments 2 and 3 which were performed in late September, the soil presented less surface disturbance due to the cumulative action of rainfall within that period (Papanicolaou et al., 2015b). Therefore, despite tamping with plywood, remnants of tillage effects remained in Experiment 1 yielding different initial microroughness length scales than Experiments 2 and 3. All cases, however, exhibited initial microroughness length less than 2 mm corresponding to smooth surface bed conditions as confirmed with the laser scanner. Dry soil bulk density was 1.25 g/cm$^3$ for Experiment 1, and about 6% higher for Experiments 2 and 3 due to self-weigh consolidation of soil.

Figure 4a provides an example of the experimental plot at pre-rainfall and post-rainfall conditions. Since the focus of this research is only on plot regions where raindrop detachment is dominant over runoff, we are using the scanned profiles that correspond only to these upslope locations, which are shown in Fig. 4b. No rill formation ever took place in these regions. For scanned profiles within the Region of Interest (ROI) (i.e., a selected 200 mm x 200 mm window size), we extracted the data for further statistical and geostatistical analyses by utilizing the public domain R software (https://www.r-project.org/). The geostatistics ('gstat') and spatial analysis ('sp') libraries were imported to create sample semivariograms.

## 2.2 Soil Surface Roughness Quantification

The RR index calculated from Eq. (1) was used in this study as the principal method to quantify soil surface roughness due to its frequent and widespread use in various studies and landscape models as a descriptor of microroughness. The RR index, however, requires that there is no spatial correlation between the surface elevations (Huang and Bradford, 1992). If correlation exists within a certain spatial scale, RR will likely change with the changing window size of observed data (Paz-

Ferreiro et al., 2008) and may be dependent on the resolution of the measurement device (Huang and Bradford, 1992). Thus, alternative scale-independent methods that consider spatial correlation have been developed by other researchers in order to address this issue. These methods include first-order variogram analysis (Linden and van Doren, 1986; Paz-Ferreiro et al., 2008), semivariogram analysis (Vázquez et al., 2005; Oleschko et al., 2008; Rosa et al. 2012; Vermang et al., 2013), fractal

models based on Fractional Brownian Motion (Burrough, 1983a; Vázquez et al., 2005; Papanicolaou et al., 2012; Vermang et al., 2013), multifractal analysis (Lovejoy and Schertzer, 2007; Vázquez et al., 2008), Markov-Gaussian model (Huang and Bradford, 1992; Vermang et al., 2013), and two-dimensional Fourier Transform (Cheng et al., 2012), among others. We herein employ indices derived from the first-order variogram and the semivariogram as alternatives to the RR index. These include the crossover length, the Markov-Gaussian variance length scale, and the limiting difference.

The crossover length derived from semivariogram analysis is an index that is commonly used in most recent soil microrelief studies to describe surface microroughness. It has the advantage of its quantification being scale independent through the consideration of the spatial correlation between surface elevations (Vázquez et al., 2007; Paz-Ferreiro et al., 2008; Tarquis et al., 2008). The semivariogram is calculated from the following equation:

$$\gamma(h) = \frac{1}{2n(h)} \sum_{i=1}^{n(h)} [Z(x_i + h) - Z(x_i)]^2 , \qquad\qquad (2)$$

where $\gamma(h)$ is the semivariance, $h$ is the lag-distance between data points, $Z(x)$ is the elevation height value at location $x$ and $n(h)$ is the total number of pairs separated by lag-distance $h$ considered in the calculation. The semivariogram is the plot of the semivariance with respect to the lag-distance.

Key indices for describing soil surface roughness can be derived from the semivariogram. Assuming a fractional Brownian motion model for describing soil surface roughness, as proposed in the pioneering work of Mandelbrot and van Ness (1968),

the following expression for $\gamma(h)$ that incorporates the generalized Hurst exponent, $H$ is obtained (Huang and Bradford, 1992; Vázquez et al., 2007; Paz-Ferreiro et al., 2008; Tarquis et al., 2008):

$$\gamma(h) = l^{2-2H} h^{2H}, \qquad\qquad (3)$$

where $H$ is a measure of the degree of correlation between the surface elevations at lag distance $h$ with $0 < H < 1$ and $l$ is the crossover length. The crossover length is a measure of the vertical variability of soil surface roughness at the particular

scale where the fractal dimension is estimated, hence greater roughness is associated with larger crossover length values and vice versa (Huang and Bradford, 1992). The generalized Hurst exponent is a less sensitive descriptor of soil surface evolution as influenced by rainfall (Vázquez et al., 2005), hence attention is mostly centered on the crossover length. Given the semivariogram plot calculated using Eq. (2), $H$ and $l$ can be extracted by fitting a power law relationship in the form of $y = Ax^B$ to the semivariance-lag distance data, where $y = \gamma(h)$ and $x = h$. According to Eq. (3), the $B$ regression variable

gives the generalized Hurst exponent value and the $A$ regression variable yields the crossover length.

The Markov-Gaussian model is a random process that has been adopted for the quantification of soil surface roughness (Huang and Bradford, 1992; Vermang et al., 2013). In that case, the semivariogram is written as an exponential-type function with the following form:

$$\gamma(h) = \sigma^2\left(1 - e^{-h/L}\right), \tag{4}$$

where $\sigma$ is the variance length scale, representing the roughness of a surface at the large scale, and $L$ is the correlation length scale, which is a measure of the rate at which small scale roughness variations approach the constant value of $\sigma$. These indices are obtained by fitting the exponential-type function of Eq. (4) to the semivariogram obtained from Eq. (2).

Finally, the limiting difference (LD) index is another index adopted to quantify soil surface roughness. It is calculated from the first-order variogram (Linden and van Doren, 1986; Paz-Ferreiro et al., 2008), which is written in the form:

$$\Delta Z(h) = \frac{1}{n(h)}\sum_{i=1}^{n(h)}|Z(x_i + h) - Z(x_i)|, \tag{5}$$

Then, a linear relationship is fitted between $1/\Delta Z(h)$ and $1/h$:

$$1/\Delta Z(h) = a + b/h, \tag{6}$$

The limiting difference (LD) index is then calculated as $LD = 1/a$. $LD$ has units of length, and represents the value of the first-order variance at large lag distances. It is considered as an indicator of soil surface roughness, thus adopted in the present study as an additional roughness index.

In order to negate the effects of the differences found in initial microrelief amongst the three runs and compare rainfall induced changes in relative terms, the results from the rainfall experiments are presented in the form of ratios of the

roughness indices. More precisely, the RR ratio, defined as the ratio of the RR index post-rainfall over the RR index prior to the rainfall ($RR_{post}/RR_{pre}$), is calculated for each experiment. Semivariograms are plotted under pre- and post-rainfall conditions at the ROI to assess the spatial correlation of surface elevations. Along the same lines, ratios between pre- and post-rainfall conditions are calculated for the crossover length, the variance length scale of the Markov-Gaussian model, and the limiting difference to assess changes in microroughness along with the RR ratio.

**3. Results**

**3.1 Changes in the RR index**

Based on visual inspection of the DEMs in Fig. 4b, it is evident that roughness at the upslope regions increases with rainfall. Figure 5 shows the RR ratio, i.e., $RR_{post}/RR_{pre}$, with respect to the initial value of RR for the present study along with other studies that quantify rainfall induced microroughness changes. The dashed line at the RR ratio value of unity reflects no

change in roughness, thus all points above that line show an increasing trend with rainfall, while all points below show a decreasing trend with rainfall. All the studies capture a wide range of initial RR values – up to 21 mm – and it is clear that our study captures the behavior of RR for an initial range that was not covered before. Figure 5 suggests that roughness may increase with raindrop impact for a range of low initial RR values (< 5 mm), while it consistently decays for high initial RR values (> 5 mm). It is acknowledged that the values of the roughness indices among different studies may reflect different

conditions such as rainfall forcing and soil type. For example, Vázquez et al. (2007) used clay textured soil, Vázquez et al.

(2008) used silt loam textured soil, while our study along with all the other studies cited conducted rainfall experiments for silty clay loam textured soil. Rainfall intensities and cumulative rainfall amounts varied significantly among studies.

Table 1 summarizes the results of this study along with results from the selected studies in quantitative terms, documenting the RR index values before and after the rainfall events, the cumulative rainfall, as well as the associated RR ratio. Two inferences can be made from Table 1. First, our study along with Vázquez et al. (2008) and Zheng et al. (2014), which were performed for the smooth surface initial condition, report an increase in RR with rainfall in general. Exception seems to hold for one soil surface of the study of Vázquez et al. (2008), as well as the smooth surfaces of Vermang et al. (2013) which show decaying roughness due to rainfall because of different soil type and rainfall conditions. Second, the present study indicates that the RR ratio becomes higher with higher rainfall intensity when the surface is classified as smooth, whereas the opposite tends to hold for soil surfaces classified as disturbed (Fig. 5, Table 1). Vázquez et al. (2008) and Zheng et al. (2014) recorded an increase in RR with rainfall but had significantly lower values of RR ratio than we did. This may be attributed to the fact that they either applied lower rainfall intensity or the initial microroughness conditions in their study were higher. Other studies not included in Table 1 have also shown increasing trends of roughness with rainfall, as quantified with the use of different indices. For instance, Huang and Bradford (1992) calculated the semivariograms for different surfaces and used fractal and Markov-Gaussian parameters to quantify the roughness. Markov-Gaussian analysis showed a relative increase in the roughness parameter for a surface of low initial roughness. Finally, Rosa et al. (2012) introduced the Roughness Index which is estimated from the semivariogram sill in order to quantify roughness, and observed its increase with rainfall under low initial roughness conditions. That increase was attributed to the fragmentation of aggregates and clods to smaller aggregates but was not linked to smooth bare soil surface conditions.

The results outlined above for the use of the RR index as a descriptor of change in microroughness have been based on the assumption that there is no statistically significant spatial correlation in elevation readings between neighboring locations at the ROI, so they are valid only under this assumption. The following subsection outlines and discusses the results of the semivariogram analysis and additional indices in order to confirm this assumption and compare with the RR index method.

## 3.2 Changes in alternative roughness indices

Semivariograms and first-order variograms were obtained from geostatistical analysis and plotted at four different angles – 0°, 45°, 90°, and 135°– with respect to the downslope direction Since the action of rainfall is isotropic and adds no systematic trend along any direction, no significant differences were expected between semivariograms. A nonparametric test for spatial isotropy was performed per Guan et al. (2004) using the public domain R statistical package with the 'spTest' library. The spatial isotropy hypothesis was confirmed ($p < 0.05$). Thus, there would be no bias in taking any direction to calculate the semivariograms and the associated crossover lengths.

The semivariograms calculated at the ROI were chosen to be in the downslope direction at an angle of 0° and are presented for each experiment in Fig. 6. The vertical dashed lines designate the lag distances above which the spatial autocorrelation of the elevations is not statistically significant. These lag distances are approximately 10 mm, so the selected 200 mm window

size of the ROI is almost 20 times greater than the spatial autocorrelation range. This implies that the window size of the ROI falls at the scale of the semivariogram sill (which is defined as the near-constant value of semivariance at large lag distances where the semivariogram levels out – see horizontal dashed lines in Fig. 6). RR is directly related to the semivariogram sill (e.g., Vázquez et al., 2005; Vermang et al., 2013), therefore it can be considered independent of the selected window size, given that the latter far exceeds the spatial autocorrelation range.

Fig. 6 shows that the post-rainfall sills are greater than their corresponding pre-rainfall values. Also, the difference in sills between pre- and post-rainfall conditions for the 30 mm/h precipitation intensity is much lower than those of the 60 mm/h and 75 mm/h events. These observations are in accordance with visual inspection of the surfaces as well as with the results noted earlier for the RR ratio (see Fig. 5, Table 1). Complete agreement between the trends of the RR index, the semivariogram sill, and visual inspection of the surfaces justify the use of the RR index as a representative and unbiased descriptor of microroughness.

Table 2 summarizes the results of this study along with results from other selected studies in quantitative terms, documenting the crossover length values before and after the rainfall events, the cumulative rainfall, as well as the associated crossover length ratio. The final roughness state of the bed surface after the application of rainfall (for the studies considered herein), tends to have a crossover length in the range of 0.2-4 mm. It is seen that the existing studies with initial disturbed surface conditions report a decrease in the crossover length after rainfall, contrary to our study where we observe an increase for the initial smooth conditions (e.g., Vázquez et al., 2007; Paz-Ferreiro et al., 2008; Vermang et al., 2013). Crossover length ratios greater than unity reflect an increase of soil surface roughness with rainfall. Similar to the RR ratio, the crossover length ratio is greater at the high precipitation intensity cases (60 and 75 mm/h) than at the low precipitation intensity case (30 mm/h).

Table 3 lists the Markov-Gaussian variance length scale and the limiting difference indices for the three experimental tests, and their relative change after the rainfall. These indices show an increase with rainfall that is of the same magnitude and trend as the RR index and crossover length, and provide a supplemental analysis about the role of rainfall intensities on the relative increase in roughness. Our findings were compared against those reported in the literature. Huang and Bradford (1992) studied the evolution of soil surface roughness with the Markov-Gaussian variance length scale, and saw an increase of 6% in roughness for a surface of low initial roughness. Moreover, Paz-Ferreiro et al. (2008), who used the LD index to quantify soil surface roughness, also recorded a 10% increase in the LD index for a low roughness conventional tillage soil surface. The higher relative increase in roughness seen in our study (Table 3) compared to other studies is attributed to the significantly lower initial roughness conditions in addition to different soil types and management.

Overall, the results provided suggest that all the indices employed in this study may be used interchangeably to characterize rainfall induced changes in soil surface roughness, and can capture an increase in soil surface roughness, especially for low microroughness scales on the order of 2-5 mm. For these microroughness scales, the relative increase in roughness is also shown to increase with rainfall intensity.

## 4. Discussion and Conclusions

Few studies have been developed to assess microscale variation under controlled conditions to purposely examine increase in RR with rainfall intensity. Unique experiments are presented herein that were designed to help us decipher the role of rainsplash on increasing RR by isolating the role of other processes such as runoff, variable water content, bare soil surface, soil texture, etc. Our findings suggest the existence of a characteristic roughness scale in the magnitude of 2-5 mm below which RR is expected to increase and above which RR is expected to decrease due to the action of rainfall. It is further demonstrated that for low microroughness scales the relative increase in roughness increases with rainfall intensity. Another significant outcome of this study is the fact that the mere action of rainfall cannot completely smoothen out a bed soil surface, thereby localized microroughness residuals will always remain at the locations where the action of runoff is low or absent. Increase in microroughness further infers increase in depression storage at the soil surface prior to runoff generation (Kamphorst et al., 2000), which can affect ponding and flow pathway patterns especially at the onset of a storm event (Onstad, 1984). The results obtained are consistent with findings of other studies that have examined length scales up to 5 mm. These length scales (i.e., ~2-5 mm) have been found to be common in agricultural landscapes that are subject to prolonged exposure to rainfall impact, runoff, and freeze-thaw cycles (Burwell et al., 1963; Allmaras et al., 1966; Burwell et al., 1969; Cogo, 1981; Currence and Lovely, 1970; Steichen, 1984; Unger, 1984; Zobeck and Onstad, 1987). Within these landscapes, the reported increase is expected to occur during the early part of the storm where rainsplash action may be more important than runoff. The exact mechanisms leading to increase in roughness are unknown. Changes in roughness during a storm event can be attributed to compression and drag forces from the raindrop impact on the soil, angular displacement due to rainsplash, aggregate fragmentation, and differential swelling (Al-Durrah and Bradford, 1982; Warrington et al., 2009; Rosa et al., 2012; Fu et al., 2016). It is recognized that dryer, silty type soils may not exhibit the increase in RR shown here. Also, the role of sealing may be important on roughness development under bare soil conditions and needs further examination. Soil water retention characteristics of the soils under sealing and its implication to RR must be considered (Saxton and Rawls, 2006). Additionally, regions exhibiting different median raindrop diameters may experience different soil surface roughness evolution due to different aggregate fragmentation and rain splash effects (Warrington et al., 2009; Rosa et al., 2012; Fu et al., 2016).

Our findings provide a better understanding of the highly dynamic phenomenon of soil surface microroughness evolution under the impact of rainfall. This study motivates further research on the extent of influence of the examined phenomenon and its mathematical formulation for modeling applications. For instance, current modeling tools of soil surface processes may predict a total decay of soil surface roughness after subsequent rainfall events (e.g., Potter, 1990) which may not be the case. Finally, this study and other studies demonstrate that the evolution of soil surface roughness in response to rainfall is dependent on initial roughness conditions and can contribute to hydrology, i.e., another factor shaping the soil surface (e.g., through runoff). Different behavior of surface roughness evolution, i.e., increase or decrease, depending on initial roughness conditions indicates a dynamic and nonlinear feedback between hydrologic response and surface roughness which may

affect depression storage, ponding and flow pathways (Kamphorst et al., 2000; Gómez and Nearing, 2005). However, the extent to which soil surface roughness increase would affect depression storage, ponding, and flow pathways is unknown, and further research to quantify this effect is needed. Nonetheless, the current findings may help explain some modeling discrepancies in terms of depression storage and runoff predictions.

On an annual basis, Abaci and Papanicolaou (2009) and Abban et al. (2016) highlight the importance of the seasonal variation of land cover on sediment output in agricultural Intensively Managed Landscapes (IMLs), indicating that during certain periods, the combination of high magnitude events and bare soil will severely increase erosion. This point is of relevance here given the soil surface in agricultural IMLs is bare 30-75% of the time during the calendar year. Models simulating these periods at the microscale are likely to be sensitive to the treatment (and definition) of the soil surface

microroughness, and thus, require an adequate determination of the soil surface roughness length scales for accurately modeling the hydrologic response of hillslopes. To the extent that microscale processes are considered significant, we argue that such models should adequately capture the increasing and decreasing trends in soil microroughness during all stages of a storm event in order to accurately predict local response to rainfall. The extent to which the increase in RR recorded herein can affect erosion processes is not yet known. However, it has been noted that different values of RR can affect flow

pathways and runoff, which consequently can affect erosion processes (Gómez and Nearing, 2005).

The majority of existing models assume that RR always decays over time with rainfall. Few models consider the reverse condition where the soil surface is initially smooth as defined in the current paper and RR increases under the action of raindrop. By providing the ratios of increase in roughness indices with rainfall intensity, the parameterization of the evolution of surface roughness with rainfall could be improved for current models. Future research will provide a better

understanding of the extent to which the initial increase in roughness in the early part of the storm could have an impact on flow pathways, runoff, and processes at subsequent parts of the storm.

**Competing interests**

The authors declare that they have no conflict of interest.

**Acknowledgments**

The present study was in parts supported by the National Science Foundation grant EAR1331906 for the Critical Zone Observatory for intensively managed landscapes (IML-CZO), which comprises multi-institutional collaborative effort. The authors, especially the corresponding author, would like to acknowledge the help provided by Dr. Chi-Hua Huang from the USDA-ARS National Soil Erosion Research Lab, West Lafayette, IN regarding the purchase of the laser system used in this research to map the RR. The fifth author was partially supported by the University of Iowa NSF IGERT program,

Geoinformatics for Environmental and Energy Modeling and Prediction. This research was supported by the NASA

EPSCoR Program [Grant #NNX10AN28A] and the Iowa Space Grant Consortium [Grant # NNX10AK63H]. Finally, the first author during part of this analysis has been supported by the USDA-AFRI grant. The data of this research are available to the interested reader upon written request to any of the first three authors.

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

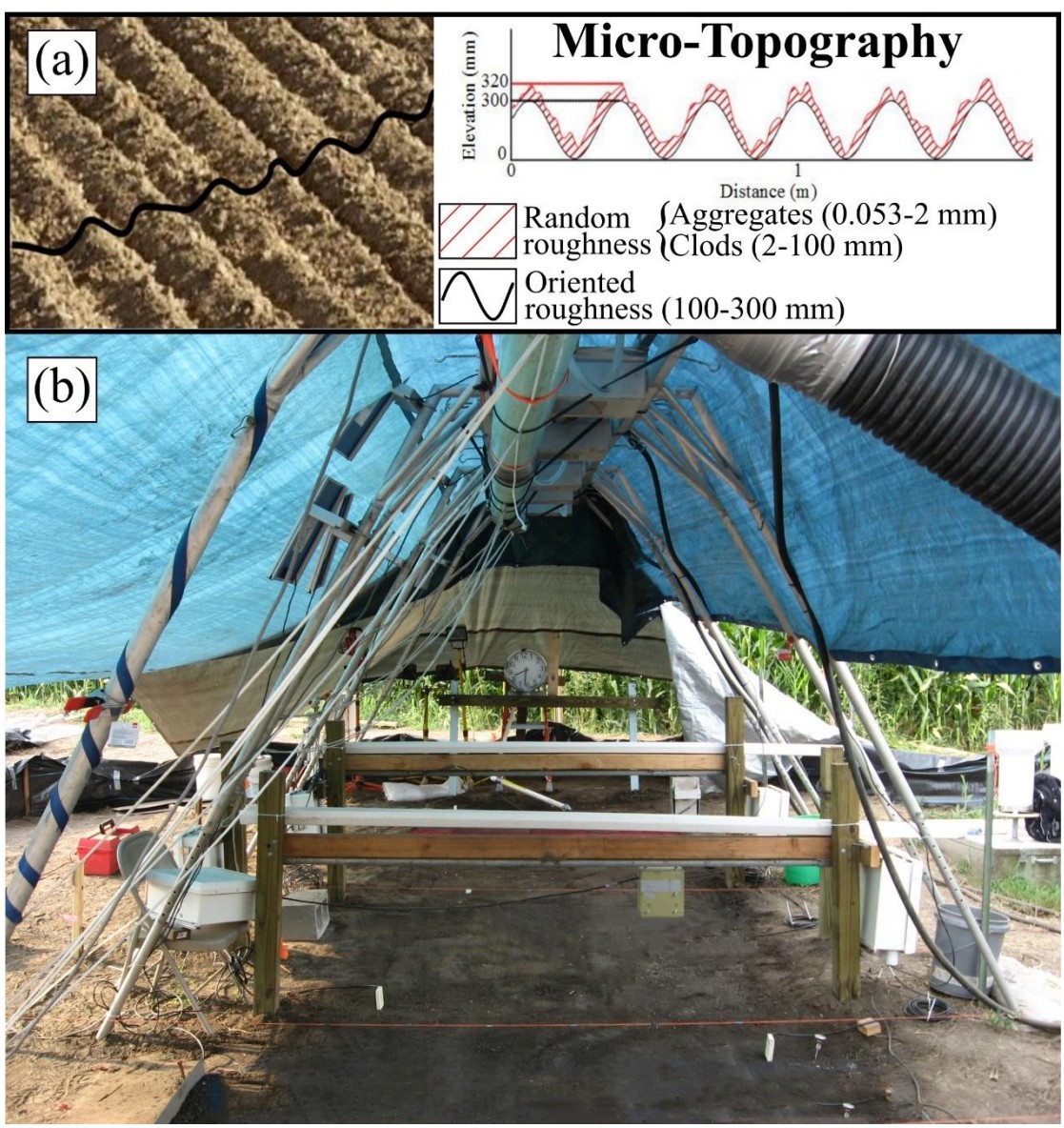

**Figure 1: (a) Types of soil surface microroughness. (b) Experimental plot. The rainfall simulator is placed above the bare soil surface and a base made of wood is put into place to facilitate the movement of the surface-profile laser scanner.**

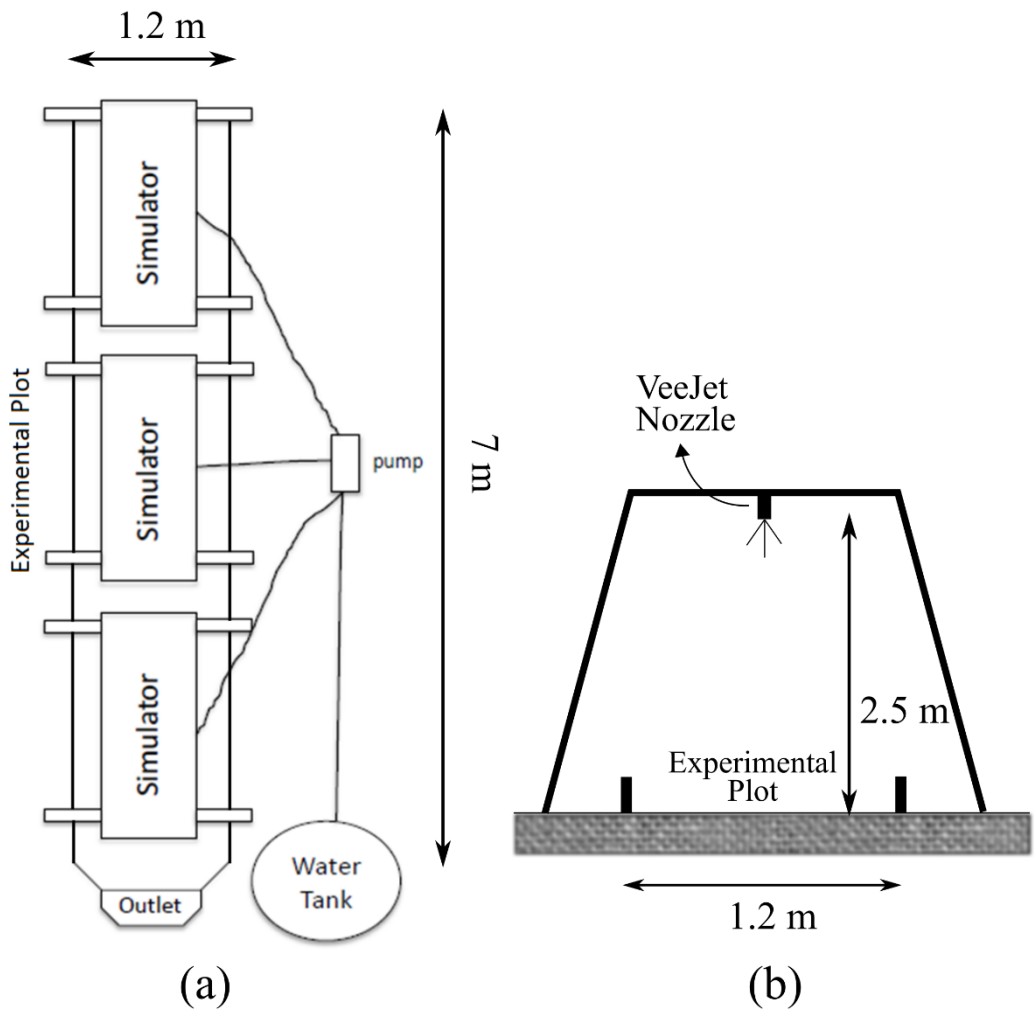

**Figure 2: Setup of the experimental tests: (a) Rainfall simulators are mounted in series and a pump provides them with water from a tank. (b) Rainfall simulators are placed and adjusted at a height of 2.5 m above the experimental plot surface to ensure drop terminal velocity is reached.**

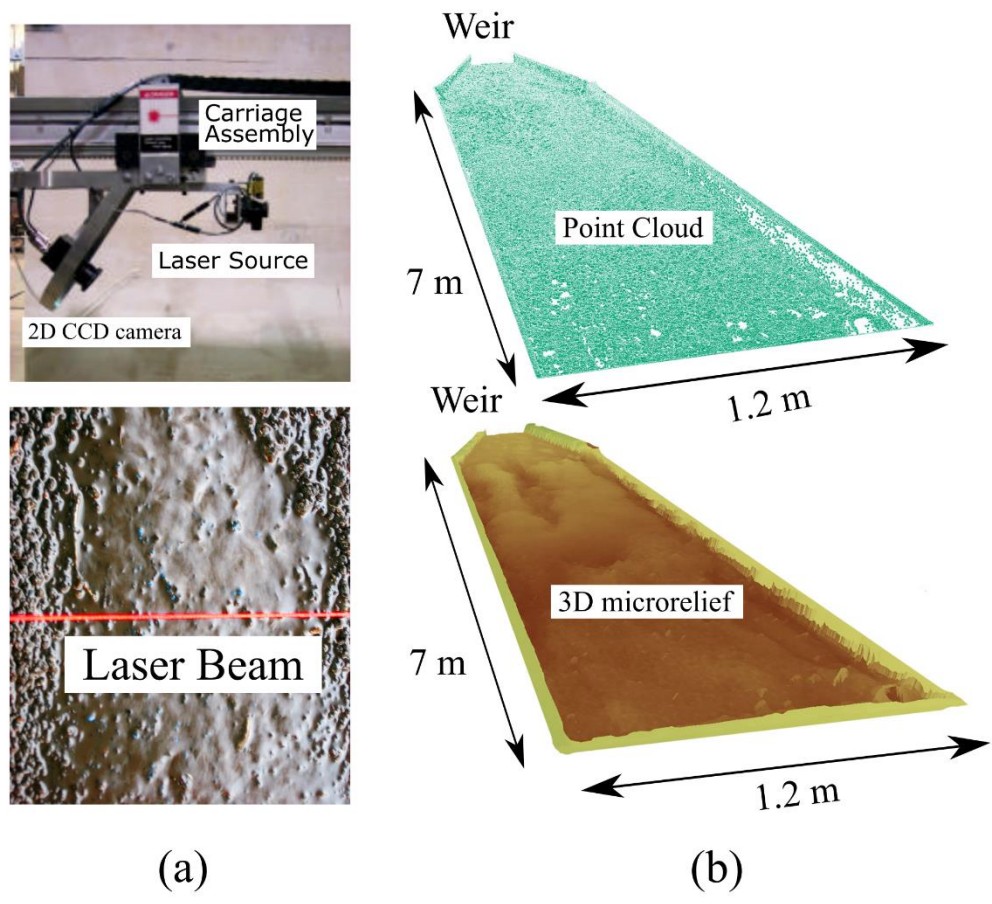

(a)             (b)

**Figure 3: (a) Instantaneous digital surface-profile laser scanner used in the experimental runs and laser beam projected on the soil surface. (b) Cloud of (x,y,z) data acquired from the laser scanner for an experimental test along with the associated 3D representation of the soil surface microrelief through inverse distance weighted interpolation .**

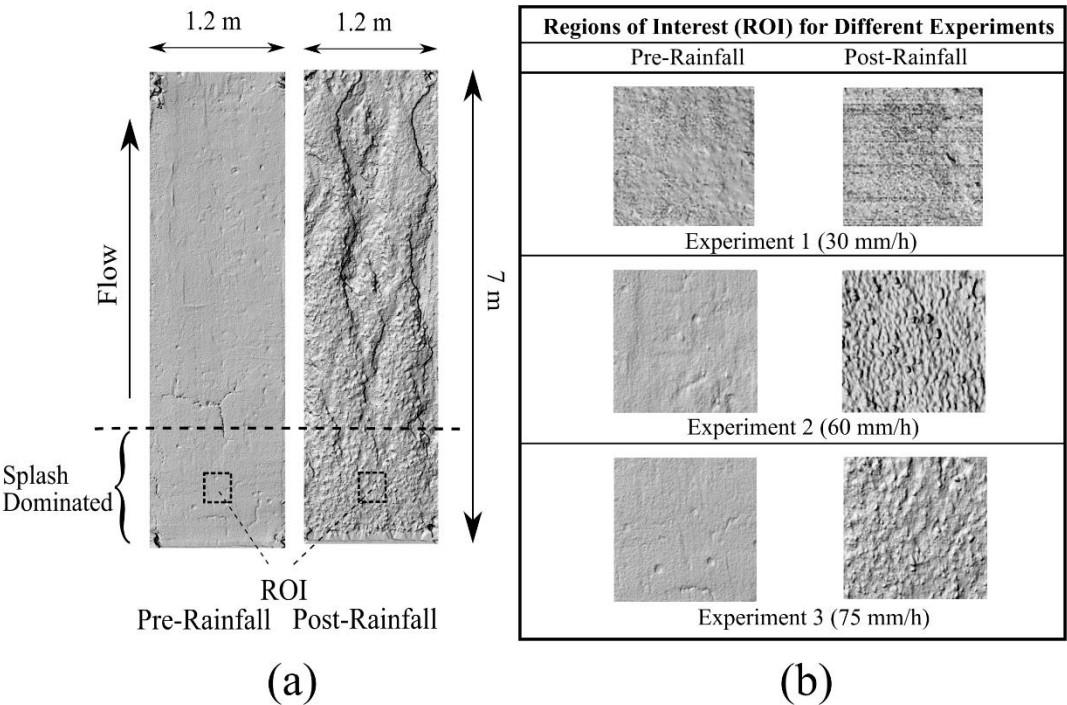

**Figure 4: (a) Experimental plot under pre- and post-rainfall conditions for an experimental test. The dashed boxes indicate the extent of the Region of Interest(ROI), where raindrop detachment is dominant over runoff. (b) Scanned profiles extracted from the laser-scanned areas of the three experimental tests considered, under both pre- and post-rainfall conditions.**

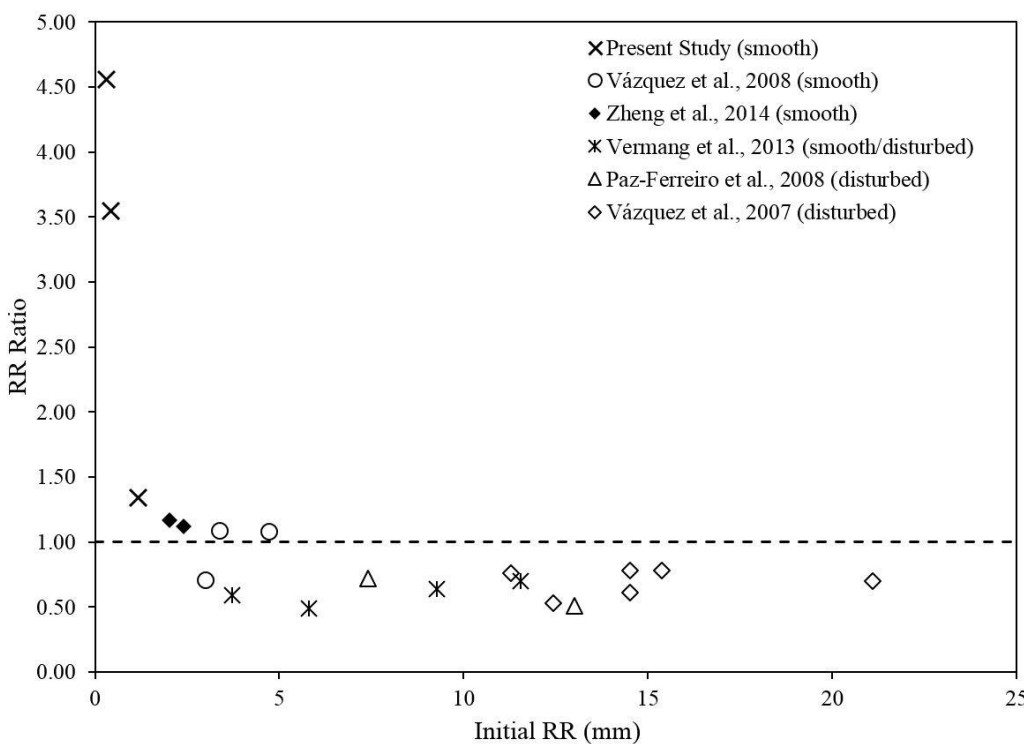

**Figure 5: Random Roughness (RR) Ratio versus initial RR for this study and other selected studies.**

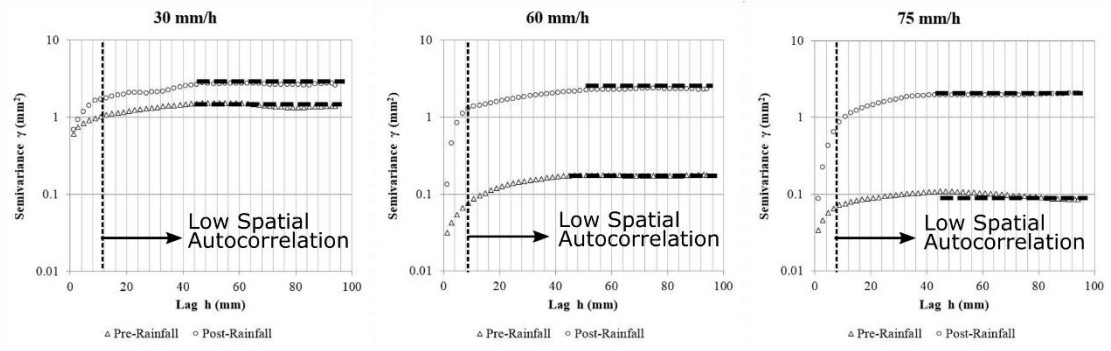

5 **Figure 6: Semivariograms at the region of interest for the three experimental tests, under pre- and post-rainfall conditions. Horizontal dashed lines indicate the semivariogram sills and vertical dashed lines indicate the lag distance above which the spatial autocorrelation of the elevations is negligible.**

**Table 1: Summary of the rainfall induced change in the RR index in the experimental tests of this study, as well as in experiments reported in the literature. Smooth conditions refer to initial microroughness on the order of 2-5 mm and disturbed conditions refer to initial microroughness greater than 5 mm. Cumulative rainfall amounts are also provided.**

| | Cumlative rainfall (mm) | Pre-rainfall RR (mm) | Post-rainfall RR (mm) | RR Ratio |
|---|---|---|---|---|
| Present Study (smooth) | | | | |
| 30 mm/h | 150 | 1.17 | 1.57 | 1.34 |
| 60 mm/h | 300 | 0.42 | 1.48 | 3.55 |
| 75 mm/h | 375 | 0.32 | 1.46 | 4.56 |
| Vázquez et al., 2008 (smooth) | | | | |
| Soil MA4 | 85 | 3.39 | 3.70 | 1.09 |
| Soil MA6 | 50 | 3.00 | 2.13 | 0.71 |
| Soil LU1 | 195 | 4.72 | 5.10 | 1.08 |
| Zheng et al., 2014 (smooth) | | | | |
| Straight Slope | ~60 | 2.01 | 2.35 | 1.17 |
| Raking Cropland | ~135 | 2.40 | 2.68 | 1.12 |
| Vermang et al., 2013 (smooth/disturbed) | | | | |
| Very Smooth | 150 | 3.71 | 2.19 | 0.59 |
| Smooth | 150 | 5.80 | 2.82 | 0.49 |
| Rough | 150 | 9.27 | 5.92 | 0.64 |
| Very Rough | 150 | 11.55 | 8.13 | 0.70 |
| Paz-Ferreiro et al., 2008 (disturbed) | | | | |
| Minumum tillage (2003) | 229 | 13.00 | 6.63 | 0.51 |
| Minimum tillage (2004) | 350 | 7.40 | 5.32 | 0.72 |
| Vázquez et al., 2007 (disturbed) | | | | |
| Disc Harrow | 233 | 15.38 | 11.99 | 0.78 |
| Disc Plow | 233 | 21.09 | 14.69 | 0.70 |
| Chisel Plow | 233 | 11.29 | 8.60 | 0.76 |
| Disk harrow + Disc Level | 295 | 12.42 | 6.58 | 0.53 |
| Disc Plow + Disc Level | 295 | 14.51 | 11.38 | 0.78 |
| Chisel Plow + Disc Level | 295 | 14.51 | 8.83 | 0.61 |

**Table 2: Summary of the rainfall induced change in the crossover length in the experimental tests of this study, as well as in experiments reported in the literature. Smooth conditions refer to initial microroughness on the order of 2-5 mm and disturbed conditions refer to initial microroughness greater than 5 mm. Cumulative rainfall amounts are also provided.**

| | Cumulative Rainfall (mm) | Pre-rainfall l (mm) | Post-rainfall l (mm) | Crossover length Ratio |
|---|---|---|---|---|
| Present Study (smooth) | | | | |
| 30 mm/h | 150 | 0.71 | 0.73 | 1.03 |
| 60 mm/h | 300 | 0.09 | 0.20 | 2.13 |
| 75 mm/h | 375 | 0.15 | 0.39 | 2.56 |
| Vermang et al., 2013 (smooth/disturbed) | | | | |
| Very Smooth | 150 | 2.07 | 1.07 | 0.52 |
| Smooth | 150 | 4.56 | 1.16 | 0.25 |
| Rough | 150 | 6.93 | 1.46 | 0.21 |
| Very Rough | 150 | 5.06 | 1.63 | 0.32 |
| Paz-Ferreiro et al., 2008 (disturbed) | | | | |
| Minimum Tillage (2003) | 229 | 4.73 | 3.80 | 0.80 |
| Minimum Tillage (2004) | 350 | 5.49 | 1.69 | 0.31 |
| Vázquez et al., 2007 (disturbed) | | | | |
| Disc Harrow | 233 | 7.69 | 2.86 | 0.37 |
| Disc Plow | 233 | 10.32 | 1.97 | 0.19 |
| Chisel Plow | 233 | 6.43 | 1.71 | 0.27 |
| Disk harrow + Disc Level | 295 | 6.69 | 1.19 | 0.18 |
| Disc Plow + Disc Level | 295 | 9.25 | 1.17 | 0.13 |
| Chisel Plow + Disc Level | 295 | 5.01 | 1.16 | 0.23 |

5  **Table 3: Summary of the rainfall induced change in the Markov-Gaussian variance length scale and limiting difference indices for the experimental tests of this study.**

| ` | Cumulative Rainfall (mm) | Pre-rainfall $\sigma$ (mm) | Post-rainfall $\sigma$ (mm) | $\sigma$ Ratio | Pre-rainfall $LD$ (mm) | Post-rainfall $LD$ (mm) | $LD$ Ratio |
|---|---|---|---|---|---|---|---|
| 30 mm/h | 150 | 1.19 | 1.63 | 1.37 | 0.79 | 0.87 | 1.10 |
| 60 mm/h | 300 | 0.42 | 1.52 | 3.62 | 0.26 | 0.87 | 3.39 |
| 75 mm/h | 375 | 0.31 | 1.43 | 4.56 | 0.15 | 0.71 | 4.84 |