# Peer review of "Quantifying the changes of soil surface microroughness due to rainfall-induced erosion on a smooth surface"

_Nonlinear Processes in Geophysics, 2016_

## Referee Comment (RC1) · Anonymous Referee #1 · 31 Jan 2017

The manuscript entitled "Quantifying the changes of soil surface microroughness due to rainfall-induced erosion on a smooth surface" (Reference number NPG-2016-76) authored by B.K.B. Abban, A.N. Papanicolau, C.P. Giannopoulos, D.C. Dermisis, K.M. Wacha, C.G. Wilson, and M. Elhakeem presents results from a simulated rainfall experiment consisting of applying three different intensities to a smoothed bare soil surface. Authors calculated two widely-used indicators of surface roughness and discussed the implications of their results for modelling approaches. The reported work is interesting and fits within the scope of Nonlinear Processes in Geophysics. However, the manuscript has an unusual organization and authors mixed methods with results and discussion. Moreover, relevant information is missing from the Materials

and Methods section. Another major concern that I have after reading this manuscript is the feeling that authors overestimated the relevance of their results and reached conclusions that are not sufficiently proven by their data, especially because of the number of events that they experimented (only three, one per rainfall intensity with no replications). Finally, a few English mistakes must be corrected. In the following lines, I provide the authors with some suggestions in order to improve their manuscript. They must correct them in order that this manuscript achieves the standard quality for being published in Nonlinear Processes in Geophysics. Therefore, I recommend the rejection of this manuscript. However, if the editor feels that the research presented is of interest, I made a great number of comments and suggestions in an appended file.

Please also note the supplement to this comment:
http://www.nonlin-processes-geophys-discuss.net/npg-2016-76/npg-2016-76-RC1-supplement.pdf

**Supplement:**

The manuscript entitled "Quantifying the changes of soil surface microroughness due to rainfall-induced erosion on a smooth surface" (Reference number NPG-2016-76) authored by B.K.B. Abban, A.N. Papanicolau, C.P. Giannopoulos, D.C. Dermisis, K.M. Wacha, C.G. Wilson, and M. Elhakeem presents results from a simulated rainfall experiment consisting of applying three different intensities to a smoothed bare soil surface. Authors calculated two widely-used indicators of surface roughness and discussed the implications of their results for modelling approaches.

The reported work is interesting and fits within the scope of Nonlinear Processes in Geophysics. However, the manuscript has an unusual organization and authors mixed methods with results and discussion. Moreover, relevant information is missing from the Materials and Methods section. Another major concern that I have after reading this manuscript is the feeling that authors overestimated the relevance of their results and reached conclusions that are not sufficiently proven by their data, especially because of the number of events that they experimented (only three, one per rainfall intensity with no replications). Finally, a few English mistakes must be corrected.

In the following lines, I provide the authors with some suggestions in order to improve their manuscript. They must correct them in order that this manuscript achieves the standard quality for being published in *Nonlinear Processes in Geophysics*.

Therefore, I recommend the rejection of this manuscript. However, if the editor feels that the research presented is of interest, I made a great number of comments and suggestions in the following pages.

**Specific comments to the authors:**

*Abstract:*

*The abstract must be greatly improved.*

*Page 1, lines 13-14: "in agricultural landscapes", this is too general and does not describe what you have done. You did not study all agricultural landscapes, even not a few of them, just one and adapted to the smooth surface conditions that you were interested in.*

*Page 1, line 17: "representative intensities", representative of what? For instance, 75 mm/h is a very high intensity; does it frequently happen in your region?*

*Page 1, line 21: I would remove "for initial microroughness length scales of 2 mm" since this is already stated in line 15 and you did not study any other length scales.*

*Page 1, line 22: How can your results contradict literature when you have said that there is no literature for the surface conditions you have assayed?*

*Page 1, line 23: "Analysis shows", what analysis?*

*Page 1, lines 23-24: This last sentence must be re-phrased and a conclusion should be added.*

*Introduction:*

*This section is well-written and provides enough information about the background of the presented work.*

*Page 1, line 29: You could remove "reported in the literature".*

*Page 2, line 4: "From the outlined above" instead of "From the classes outlined above".*

*Page 2, line 7: What is "scape"? Is this a mistake? Should it be "scale"?*

*Page 2, line 8: I am not sure if "enhancing" is the right word here.*

*Page 2, line 10: I do not understand why you cite Paz-Ferreiro et al., 2008 here. Besides, according to the reference list, Allmaras (1966) should be Allmaras et al. (1966).*

*Page 2, line 21: "Few to none"? Not sure about this.*

*Page 2, line 22: "This condition"? Do you mean less than 2 mm?*

*Page 2, lines 22-23: Here you say that initial microroughness scales less than 2 mm is the prevalent condition in agricultural hillslopes. However, I am not so sure that this is the prevalent state.*

*Page 2, lines 25-28: This portion of text about the models is not very well linked with the rest of the introduction. Moreover, you cite three studies. Huang and Bradford (1992), Rosa et al. (2012) and Zheng et al. (2014) that stated that RR increased with rainfall, but later you do not discuss your results in the light of these three studies, why?*

*Page 2, lines 29-30: This is already stated in the former paragraph.*

*Page 3, lines 3-9: This is a little bit messy from my viewpoint. Moreover, the two last sentences can be removed.*

*Materials and Methods:*

*This section lacks from essential information. Do you have any replication of each experiment? Did you perform only one set of measurements per experiment? It is not clear what geostatistical analysis has been performed.*

*Page 3, line 12: Maybe, you should give the geographical coordinates and elevation of your study area.*

*Page 3, line 14: What do you mean by "mixed, superactive"?*

*Page 3, line 18: "to the plots", how many plots?*

*Page 3, line 25: "widely accepted", by whom?*

*Page 3, lines 27-29: This statement is too strong, even though rainfall characteristics were similar, other regions may have different soil types than the one studied here. What are the potential biases you are referring to in this sentence?*

*Page 3, line 32: I would substitute "a priori the runs ensured that" for "before the runs confirmed that".*

*Page 4, line 1: This seems more like a result than materials and methods.*

*Page 4, line 3: What is CCD?*

*Page 3, line 5: I would use "by software" instead of "from the desktop, using a computer program".*

*Page 4, line 7: Please, provide the names and references for the specific software used.*

*Page 4, line 12: Remove "experimental" before "tests".*

*Page 4, lines 12-13: Change the sentence to "Rainfall intensities were respectively 30, 60 and 75 mm/h for experiments 1, 2 and 3".*

*Page 4, line 14: Is the duration of your rainfall events the same of the storms you are referring here? In fact, 75 mm/h during 5 hours means 375 mm in 5 hours, which seems too much. What is the return period of these events? I mean, are these storms really so usual?*

*Page 4, line 15: "Decagon soil moisture sensors", specify depth, how many and model.*

*Page 4, line 16: What do you mean by 35%? What is the field capacity of this soil? Please, specify.*

*Page 4, line 29: Remove "By definition".*

*Page 4, line 31: "extracted"·instead of "extract".*

*Page 4, line 32: "were" instead of "are".*

*Page 5, line 2: What is "its commonality found in the literature"?*

*Page 5, line 3: "was used" instead of "is used".*

*Page 5, lines 8-12: This is not clear to me. Did you use all these methods? At the end of the sentence you say "among others", what do you mean? Do you imply that you used more methods than those indicated here?*

*Page 5, line 14: I would split this sentence in two. Instead of "with the advantage of its quantification being scale independent", I would use a point and then "It has the advantage of being scale independent".*

*Page 5, line 16: "the semivariogram is a useful…" I do not think this sentence is really needed.*

*Page 5, lines 22-25: it seems rather peculiar that you explain what a semivariogram is but not what the Hurst exponent indicates.*

*Page 5, line 28: "and 0 < H < 1", this should come before, when you refer to H and not after the crossover length.*

*Page 5, lines 32-33: I did not understand this sentence. Please, re-phrase it.*

*Page 6, lines 1-4: Could you, please, re-phrase this paragraph? I do not understand it properly, it is a bit confusing.*

*Results:*

*Page 6, lines 6-14: This looks more like materials and methods than results.*

*Page 6, lines 12-14: Have these comparisons been performed against literature data? Did this literature provide ratios?*

*Page 6, line 20: Have these experiments been performed only once?*

*Page 6, lines 21-23: Please, re-phrase this. It seems redundant.*

*Page 7, lines 4-5: How did you check this significance? Only by stating that difference is less than 10%?*

*Page 7, line 6: "representative semivariogram", that corresponds to which angle?*

*Page 7, line 7: "semivariograms" instead of "semivariogram".*

*Page 7, lines 10-11: "which is considered sufficient to assume no spatial autocorrelation at the scale examined in this study", I do not get this; you checked all roughness for the 200 mm, so you should have accounted for lag distances less than 10 mm.*

*Page 7, lines 16-17: "pre-rainfall values" instead of "pre-rainfall value for all three intensities".*

*Page 7, line 18: I would use "events" instead of "precipitation intensities".*

*Page 7, line 19-20: Please, check English and re-phrase this sentence.*

*Page 7, line 22: Remove "in existing literature".*

*Page 7, line 24: "reported" instead of "report".*

*Page 7, line 25: "found" instead of "in the crossover length reported".*

*Conclusions and Discussion:*

*This section is very weak and should be greatly improved. Besides, it should be entitled "Discussion and Conclusions".*

*Page 8, line 2: Are you sure that these experiments are "unique and novel". I am also concerned*

*about the fact that you state that your experiments "mimic natural rainfall conditions" but you never described those natural rainfall conditions.*

*Page 8, lines 5-6: "which are confirmed as reliable descriptors of microroughness"; this is already known.*

*Page 8, lines 7-9: I have doubts about this, you only performed your experiment once and considered a small surface where raindrop detachment prevails over runoff; were the same conditions in the other studies? Did they consider only raindrop detachment?*

*Page 8, lines 11-12: What are the implications of this?*

*Page 8, line 13: "Roughness residuals infer depression storage residuals", what do you mean?*

*Page 8, lines 15-20: I am not sure about this. Your results come from a limited number of experiments and you are implying that they have a strong importance in various disciplines and applications... it seems overestimating your findings.*

*Page 8, lines 21-23: This must be further explained, I do not see your point.*

*Page 8, line 24: Remove "study's".*

*Page 8, lines 23-25: I really think that you are overestimating your results.*

*Page 8, lines 27-29: Yes, alright but is the initial roughness less than 2 mm? Besides, you indicate that Intensive Managed Landscapes have bare soil 75% of the time; it looks not very intensive...*

*Page 8, line 33: "landscape response to precipitation"; however, your study refers only to 200 mm² surface... is this not overestimating your results?*

*Page 9, line 1: Again overestimating the importance of your results. How this slight increase in RR may affect erosion processes?*

*Page 9, line 4: This needs, at least, a reference.*

*Page 9, line 5: "new statistical analyses", what statistical analyses did you perform?*

*Page 9, line 6: I am not sure about what you mean by "is present a priori".*

*Page 9, line 7: "in the current paper" instead of "in the paper".*

*Page 9, line 9: "is improved for current models", do you mean that is already done in current models?*

*Page 9, lines 11-12: "extension of the experiments in areas such as downslopes where concentrated flow and rilling are of importance" That you did not want to account in your study although you could have done in view of the surface of your experimental plot.*

*References:*

*Eight references are not cited in the text. Please, check them and also edit the reference list according to the journal guidelines.*

*Page 10, lines 17-18: Chu et al. (2012) is not cited in the text.*

*Page 10, lines 20-23: Why did you use upper-case letters for the title of these publications?*

*Page 10, line 27: Why did you use capital letters for CATENA?*

*Page 10, lines 30-31: Why did you use upper-case letters for the title of this publication?*

*Page 11, lines 1-3: Le Bissonais (2016) is not cited in the text.*

*Page 11, lines 4-5, 10 and 17-18: Why did you use upper-case letters for the title of these publications?*

*Page 11, line 23: Why did you use capital letters for SOIL ORGANIC CARBON DYNAMICS?*

*Page 11, line 28: Potter (1990), this reference does not follow the style of the journal; the year of publication should come at the end.*

*Page 11, line 29: Why did you use upper-case letters for the title of this publication?*

*Page 11, line 32: Why did you use capital letters for CATENA? Besides, Römkens et al. (2002) is not cited in the text.*

*Page 12, lines 4-5: Remove the quotation marks.*

*Page 12, lines 8-9: Vázquez et al. (2006) is not cited in the text.*

*Page 12, line 11: Remove "European Geo-sciences Union (EGU)".*

*Page 12, lines 15-17: Vázquez et al. (2010) is not cited in the text.*

*Page 12, lines 18-19 and 25-28: Why did you use upper-case letters for the title of these publications?*

*Page 12, lines 22-28: Zhao et al. (2014), Zhao et al. (2016) and Zheng et al. (2012) are not cited in the text.*

*Figures*

*Figure 1: Modify the caption to "(a) Types of soil surface microroughness. (b) Experimental plot. The rainfall simulator is placed above the bare soil surface and a base made of wood is put into place to facilitate the movement of the surface-profile laser scanner".*

*Figure 2: Modify the caption to "Setup of the experimental tests: (a) Rainfall simulators are mounted in series and a pump provides them with water from a tank; (b) rainfall simulators are placed and adjusted at a height of 2.5 m above the experimental plot surface to ensure drop terminal velocity is reached".*

*Figure 3: Indicate in the caption the interpolation technique that was used.*

*Figure 4: Why not a and b as in the former figures and you used left and right? You should define*

*ROI in the caption. Besides, why "part"? If the whole experimental plot was 7 x 1.2 m is the whole plot what you are representing in the right-hand side of the figure and not only part of it.*

*Figure 5: Remove "considered herein". I think that you do not need to include experiments 1, 2 and 3 if you indicate the rainfall intensity. Remove the border of the figure and the second decimal from the Y-axis.*

*Figure 6: Remove "Spatial" and "considered herein". Use "region of interest" instead of "ROI".*

*Figure 7: Remove "considered herein". I think that you do not need to include experiments 1, 2 and 3 if you indicate the rainfall intensity. Remove the border of the figure and the second decimal from the Y-axis.*

*Tables:*

*Table 1: Since the only difference amongst experiments was the rainfall intensity, this table can be removed and the information can be included in the text.*

*Table 2: You can use the rainfall intensities instead of the name of the experiments. Did all these authors perform their experiments on soils similar to yours? What were their rainfall intensities? I cannot see how you can compare them if they were different.*

*Table 3: The same comments as figure 2.*

---

## Referee Comment (RC2) · Anonymous Referee #2 · 22 Feb 2017

GENERAL COMMENTS This study analyses soil surface roughness evolution after a single event of simulated rainfall with three different intensities, namely 30, 60 and 75 mm/h. Two indices used to describe the magnitude of soil surface roughness indicate increasing values of this variable after rainfall addition.

In my opinion this manuscript does not contain significant results. This is because the experimental work has been limited to one rainfall event, and this is .obviously a main weakness in any study about soil surface roughness evolution (either increase or decay).. In addition, authors claim that the results are new, as they state that i) "Findings show a consistent increase in roughness under the action of rainfall for initial microroughness length scales of 2 mm" and ii) "This contradicts existing literature where a

monotonic decay of roughness of soil surfaces with rainfall is recorded for disturbed surfaces". However, please note that i) again, the increase in roughness (instead of a expected decrease) has been found only for the first event. What about successive events); results are not reported. ii) Increases in soil surface roughness after simulated surface rainfall and for disturbed soil surfaces have been previously reported (Please, see Vidal Vázquez et al., 2008. Assessing soil surface roughness decay during simulated rainfall by multifractal analysis. Nonlin. Processes Geophys., 15, 457–468). In this paper the evolution of the surface of three different soils was studied during successive events; two of the studied soils showed soil surface roughness increased after the first event (similar to your results)e second, but it decreased after the second and successive events; the third soil studied showed scarce trend to either increasing or decreasing surface roughness values following successive rainfall events.

SPECIFIC COMMENTS The obtained results should be put into context, with relevant references. This opinion is based in the fact that relevant studies about soil surface roughness, (including the previously cited Vidal Vázquez et al., 2008. Nonlin. Processes Geophys., 15, 457–468, and Kamphorst et al. 2000. Soil Science Society of America Journal 64(5): 1479.1458. By the way, these two manuscript present examples of soil surface roughness assessed by laser scanner as in your work. In the first work quoted (Vidal Vazquez et al., 2008) the magnitude of the roughness is not very different from that in your work and in the second (Kamphorst et al., 2000) several plots also are representative for conditions of rather low values of roughness.

In my opinion, adding more experimental data (successive events) would allow that this manuscript reaches international standards.In addition, any revision of this manuscript should address the following points: - Text should be ameliorated the text, which is not precise and provide a more clear presentation. Main corrections are expected in abstract, objectives and discussion and conclusion sections. - Mechanisms and reason for the increase in soil surface roughness after one event simulated rainfall. - There are also unnecessary figures, regarding the experimental setup, as the methodology

employed has been largely described before. - Soil composition and main characteristics should be also reported in the material and methods section. - Other significant roughness indices should be addresses, in addition to random roughness and cross over length.

Based on the above I recommend to the editor either major revision or rejection of this manuscript.

---

## Referee Comment (RC3) · Anonymous Referee #3 · 6 Mar 2017

In this manuscript, the authors address the effect of rainfall velocity on soil-air roughness quantified via the random roughness (RR) parameter. They showed that as rainfall velocity increased from 30 to 75 mm/hr the random roughness index increased as well, which is in contrast to those reported in the literature. Although more experimental data on support are required to have a more conclusive conclusion, the manuscript is well written and well organized and suitable for publication in the journal. However, some moderate revisions are required before publication.

Minor comments: P3L23: Could the authors address/discuss on how changes in median diameter would affect air-soil roughness?

P4L2: The authors should clearly state that with such a low resolution some rough

features with scale less than 0.5 mm have not been captured via their laser scanner. As the title indicates the authors address soil surface microroughness, while the resolution of the laser scanner is 0.5 mm. How is it possible to capture microroughness with a scanner of resolution of millimeters?

Did the authors measure infiltration rate or even saturated hydraulic conductivity of the tested soil? If so, what is the infiltration rate?

P8L4-8: The authors stated that, "Analysis of soil surface roughness in the region where raindrop detachment dominates and under initial smooth surface preconditions for three rainfall intensities shows a consistent increase in the RR index and crossover length, which are confirmed as reliable descriptors of microroughness. This increase contrasts the findings of most available literature . . ." Please provide a few references from the literature for the last statement.

Did the authors measure soil aggregate- or particle-size distribution? What is the range of particle sizes in mm?

---

## Author Comment (AC1) · 19 Mar 2017

**RESPONSE TO REVIEW COMMENTS**

We thank the reviewer for the insightful comments and suggestions. We believe that the edits in response to the comments have significantly improved the manuscript in terms of clarity, language, and structure. Our responses are provided below. **Please note that the original comments provided by the reviewer are in black letters and our responses are in blue letters**. In addition to these responses, we will provide a revised manuscript, as well as a copy with the tracked changes where all revisions were implemented.

Before beginning, we offer here a summary of our responses to the key concerns of the reviewer. More detailed responses are provided under the specific comments.

**1. Do you have any replication of each experiment? Did you perform only one set of measurements per experiment?**

> Three replicates of each rainfall intensity were performed. Repeatability was confirmed by evaluating changes in RR at specific cross-sections in the rainsplash dominated zone. It was found that on an average, the relative error of the RR ratios between replicates did not exceed 7%.

**2. Do you overestimated the relevance of your results and reach conclusions that are not sufficiently proven by their data?**

> Our discussion of the results has been adjusted to be more in line with the level of the analysis provided. Please see the specific comments below for each case.

A. When and where does microroughness matter?
Through these studies we were able to determine that microroughness really matters in these two cases: (1) when there is no cover, which is between harvest and planting; and (2) at the beginning of a storm event. We therefore offer only a small slice of the whole erosion process in this study under the controlled condition experiments.

B. Why perform controlled condition experiments?
Our experiments were designed to help us decipher the role of rainsplash on roughness by isolating it from the role of other processes such as runoff, variable water content, bare soil surface, texture, etc. Microroughness can lead to the formation of depression storage which can ultimately affect ponding and the formation of flow pathways.
* * *
*General Comments:*

**The manuscript entitled "Quantifying the changes of soil surface microroughness due to rainfall-induced erosion on a smooth surface" (Reference number NPG-2016-76) authored by B.K.B. Abban, A.N. Papanicolaou, C.P. Giannopoulos, D.C. Dermisis, K.M. Wacha, C.G. Wilson, and M. Elhakeem presents results from a simulated rainfall experiment consisting of applying three different intensities to a smoothed bare soil surface. Authors calculated two widely-used indicators of surface roughness and discussed the implications of their results for modelling approaches.**

The reported work is interesting and fits within the scope of Nonlinear Processes in Geophysics. However, the manuscript has an unusual organization and authors mixed methods with results and discussion. Moreover, relevant information is missing from the Materials and Methods section.

Another major concern that I have after reading this manuscript is the feeling that authors overestimated the relevance of their results and reached conclusions that are not sufficiently proven by their data, especially because of the number of events that they experimented (only three, one per rainfall intensity with no replications).

Finally, a few English mistakes must be corrected.

In the following lines, I provide the authors with some suggestions in order to improve their manuscript. They must correct them in order that this manuscript achieves the standard quality for being published in Nonlinear Processes in Geophysics.

Therefore, I recommend the rejection of this manuscript. However, if the editor feels that the research presented is of interest, I made a great number of comments and suggestions in the following pages.

Response:

Regarding restructuring, we have reallocated those areas identified by the reviewer as being out-of-place to the recommended spots. The specifics are provided in the comments addressed for the Materials and Methods and the Results.

We have also included more specifics regarding the methods used.

Those comments where the reviewer feels we have stretched too far in the relevance of our results have been adjusted to be more in line with the level of the analysis. Finally, the grammatical mistakes have been addressed with a thorough read following the edits.

**Specific Comments:**

**1. Abstract: The abstract must be greatly improved.**

Comment 1: Page 1, lines 13-14: "in agricultural landscapes", this is too general and does not describe what you have done. You did not study all agricultural landscapes, even not a few of them, just one and adapted to the smooth surface conditions that you were interested in.

Response:

We acknowledge that the expression "in agricultural landscapes" may be too general. The sentence (Page 1, lines 13-14) has been reworded as follows to be more precise:

*"This study examines the rainfall induced change in soil microroughness of a bare smooth soil surface in an agricultural field."*

**Comment 2: Page 1, line 17: "representative intensities", representative of what? For instance, 75 mm/h is a very high intensity; does it frequently happen in your region?**

Response:

The term "*representative intensities*" was based on the intensity recurrence intervals for Iowa using Huff and Angel (1992). The 30 mm/h event has a recurrence interval of ~0.25-yr, while the 60 mm/h event has a 20-yr recurrence interval. Finally, the 75 mm/h event has a 60-yr return period. Rainfall intensities of 75 mm/h usually appear in late May and June from convective thunderstorms in the study area, as seen from tipping bucket data near the test plot. Here, the Abstract (Page 1, lines 19-21) has been modified as follows and the citation placed in the reference list of the manuscript:

*"Three rainfall intensities of 30 mm/h, 60 mm/h and 75 mm/h were applied to a smoothened bed surface in a field plot via a rainfall simulator. These intensities represent the range from typical to extreme rainfall intensities that appear in the region of study (e.g., Huff and Angel, 1992)."*

Reference

Huff, F. A. and Angel, J. R.: Rainfall Frequency Atlas of the Midwest. Midwestern Climate Center Research Report 92-03. Champaign, IL, 1992.

**Comment 3: Page 1, line 21: I would remove "for initial microroughness length scales of 2 mm" since this is already stated in line 15 and you did not study any other length scales.**

Response:

The phrase "*for initial microroughness length scales of 2 mm*" was removed.

**Comment 4: Page 1, line 22: How can your results contradict literature when you have said that there is no literature for the surface conditions you have assayed?**

Response:

We understand the reviewer's confusion here. What we meant to convey was that the results contradict the commonly adopted belief in the literature that microroughness decays with rainfall regardless of the initial roughness conditions (e.g., bare flat soil vs. bare disturbed soil). We modified the sentence (Page 1, lines 25-26) as follows:

*"This contradicts the commonly adopted notion in existing literature that a monotonic decay of soil surface roughness with rainfall is expected regardless of initial surface roughness conditions."*

**Comment 5: Page 1, lines 23-24: "Analysis shows", what analysis? This last sentence must be re-phrased and a conclusion should be added.**

Response:

This last part of the abstract (Page 1, lines 23-28) has been modified to include a summary statement as follows:

*"Findings show a consistent increase in roughness under the action of rainfall, with higher rainfall intensities resulting in higher relative roughness increase. This contradicts the commonly adopted notion in existing literature that a monotonic decay of soil surface roughness with rainfall is expected regardless of initial surface roughness conditions. This study sheds light on the interaction between rainfall and smooth bare surfaces which can result in increasing soil surface roughness, and highlights the need for further examination of the phenomenon and its potential implications on hydrologic response."*

**Introduction: This section is well-written and provides enough information about the background of the presented work.**

Response:

Thank you.

**Comment 1: Page 1, line 29: You could remove "reported in the literature"**

Response:

The phrase "*reported in the literature*" was removed.

**Comment 2: Page 2, line 4: "From the outlined above" instead of "From the classes outlined above".**

Response:

This change was made.

**Comment 3: Page 2, line 7: What is "scape"? Is this a mistake? Should it be "scale"?**

Response:

The term "*soil scape*" is used often in soil science, referring to a soil column.  We have kept this term in the text.

See for example: Yaalon, D.H.: "The changing model of soil" revisited. Soil Science Society of America Journal, 76, 766-778, 2012.

**Comment 4: Page 2, line 8: I am not sure if "enhancing" is the right word here.**

Response:

The sentence was modified as follows to remove the term "*enhancing*":

"*As a result, RR affects key hydrologic processes at the soil scape and ultimately at the hillslope scale (e.g., overland flow, infiltration), by affecting the depression storage and the associated runoff and erosion processes (Gomez and Nearing, 2005; Chi et al., 2012).*"

**Comment 5: Page 2, line 10: I do not understand why you cite Paz-Ferreiro et al., 2008 here. Besides, according to the reference list, Allmaras (1966) should be Allmaras et al. (1966).**

Response:

We cite Paz-Ferreiro et al., (2008) to support the claim that RR is the most widely used descriptor of soil surface roughness.  We have also corrected the citation to "Allmaras et al. (1966)".  The text now reads as (Page 2, lines 15-16):

"*According to Paz-Ferreiro et al. (2008), the RR index, which was first proposed by Allmaras et al. (1966), is the most widely used statistical microrelief index for the evaluation of soil surface roughness.*"

**Comment 5: Page 2, line 21: "Few to none"? Not sure about this.**

Response:

We have performed an exhaustive literature search of studies that involve the quantification of soil surface roughness. Few of the existing microroughness scale studies explicitly examine the interaction of rainfall with bare soil surfaces when the surface is initially smooth and undisturbed (e.g., in most studies the surfaces are either partially covered by vegetation or disturbed by tillage). We modified the sentence (Page 2, lines 28-29) to convey the above as follows:

*"Few existing studies, to the best of our knowledge, have explicitly examined the interaction of raindrop impact with bare soil surfaces for initial microroughness scales on the order of 2-5 mm."*

**Comments 6-7: Page 2, lines 22-23: "This condition"? Do you mean less than 2 mm? Here you say that initial microroughness scales less than 2 mm is the prevalent condition in agricultural hillslopes. However, I am not so sure that this is the prevalent state.**

Response:

You are right, the sentence needed clarification. Our study explicitly deals with surfaces of initial roughness less than 2 mm. However, it is considered as an extreme condition and findings can be applicable to surfaces of the order of ~2-5 mm, which have been found to be common in agricultural landscapes according to pertinent studies (e.g., (Burwell et al., 1963; Allmaras et al., 1966; Burwell et al., 1969; Cogo, 1981; Currence and Lovely, 1970; Steichen, 1984; Unger, 1984; Zobeck and Onstad, 1987). The sentence was reworded as follows for more clarity (Pages 2-3, lines 29-2):

*"Surfaces with microroughness on the order of ~2-5 mm are common in agricultural landscapes (Burwell et al., 1963; Allmaras et al., 1966; Burwell et al., 1969; Cogo, 1981; Currence and Lovely, 1970; Steichen, 1984; Unger, 1984; Zobeck and Onstad, 1987) where the soil is "smoothened" due to long, undisturbed exposure to rainfall impact and runoff (Abaci and Papanicolaou, 2009) and freeze-thaw cycles (Zobeck and Onstad, 1987). Within these landscapes, soil surface conditions are usually bare in the period of the crop rotation between post-harvest and before plant growth is established, which approximately corresponds to 30-75% of the cyclic crop rotation period."*

**Comment 8: Page 2, lines 25-28: This portion of text about the models is not very well linked with the rest of the introduction. Moreover, you cite three studies. Huang and Bradford (1992), Rosa et al. (2012) and Zheng et al. (2014) that stated that RR increased with rainfall, but later you do not discuss your results in the light of these three studies, why?**

Response:

The studies of Huang and Bradford (1992), Rosa et al. (2012), Vázquez et al. (2008), and Zheng et al. (2014) provide quantitative indications that roughness can increase with rainfall, but they neither explicitly acknowledge the increasing trend nor link it to bare smooth surface conditions. Our goal herein is to highlight that phenomenon.

The part of the Introduction referring to these studies (Page 3, lines 3-13) was modified to outline their relevant findings:

*"There are some quantitative indications that under bare smooth surface conditions, soil surface roughness may actually increase under the action of rainfall. Specifically, the study by Huang and Bradford (1992) calculated the semivariance with respect to length scale before and after*

*rainfall, and a slight increase in roughness with rainfall was denoted using the Markov-Gaussian model for a surface with low initial roughness. Rosa et al. (2012) introduced an index (called Roughness Index) estimated from the semivariogram to describe roughness, and the index increase with rainfall under specific conditions was observed and attributed to the fragmentation of aggregates and clods to smaller aggregates. Zheng et al. (2014) reported an increase in values of the RR after the application of rainfall on smooth soil surfaces. Finally, Vázquez et al. (2008) examined the evolution of the surface of three different soils during successive events. They reported that for two out of three soils, roughness increased for the first event, however decreased for the following events; the third soil showed scarce trend to either increasing or decreasing roughness due to successive rainfall events. Nevertheless, none of the above studies explicitly stated or acknowledged the increasing trend of roughness and its potential linkage to smooth bare soil surface conditions."*

We have also added the Results further comparison between these studies and our own findings (Page 8, lines 5-19):

*"First, our study along with Vázquez et al. (2008) and Zheng et al. (2014), which were performed for the smooth surface initial condition, report an increase in RR with rainfall i**n* *general. Exception seems to hold for one soil surface of the study of Vázquez et al. (2008), as well as the smooth surfaces of Vermang et al. (2013) which show decaying roughness due to rainfall due to different soil type and conditions. Second, the present study indicates that the RR ratio becomes higher with higher cumulative rainfall amount when the surface is classified as smooth, whereas the opposite tends to hold for soil surfaces classified as disturbed (Fig. 5, Table 1). Vázquez et al. (2008) and Zheng et al. (2014) recorded an increase in RR with rainfall but had significantly lower values of RR ratio than we did. However, this may be attributed to the fact that they applied lower cumulative rainfall amount, and the initial microroughness conditions in their study were higher. Other studies that are not included in Table 1 have also shown increasing trends of roughness with rainfall, as quantified by the use of different indices. For instance, Huang and Bradford (1992) calculated the semivariograms for different surfaces and used fractal and Markov-Gaussian parameters to quantify the roughness. Markov-Gaussian analysis showed a relative increase in the roughness parameter for a surface of low initial roughness. Finally, Rosa et al. (2012) introduced the Roughness Index which is estimated from the semivariogram sill in order to quantify roughness, and observed its increase with rainfall under low initial roughness conditions. That increase was attributed to the fragmentation of aggregates and clods to smaller aggregates but was not linked to smooth bare soil surface conditions."*

**Comment 9: Page 2, lines 29-30: This is already stated in the former paragraph.**

Response:

The sentence was removed.

**Comment 10: Page 3, lines 3-9: This is a little bit messy from my viewpoint. Moreover, the two last sentences can be removed.**

Response:

To clean the text, we shortened and reorganized the last part of the introduction. We highlight more the specific objectives (Page 3, lines 20-24), which read as follows:

*"The key specific objectives of this study are (i) to quantify the soil surface microroughness of smooth bare soil surfaces before and after the effect of rainfall, and (ii) calculate the relative change in roughness for different intensities. To meet the two specific objectives we employ four commonly used indices, the RR index, the crossover length, the variance scale from the Markov-Gaussian model, and the limiting difference. The last three indices are alternate methods and used here to supplement the RR index analysis for relative change in roughness."*

**2. Materials and Methods: This section lacks from essential information. Do you have any replication of each experiment? Did you perform only one set of measurements per experiment? It is not clear what geostatistical analysis has been performed.**

Response:

All tests were performed three times to ensure repeatability in terms of homogeneity in the raindrop distribution over the rainfed test area, steady state conditions in terms of runoff volumes, and the same soil water content at the inception of the tests. Figure 1b provides a glimpse of the complex set-up needed for the experiments and hence the level of work in these experiments. Raindrop distribution was tested with image analysis (Image J software) of rain splashes within painted areas in the plot to discern them from the rest of the soil background. A weir at the outlet of the plot ensured the occurrence of steady state conditions. The continuous monitoring of volumetric soil water content showed that we had the same water content at the start of the test. Agreement between the test replicates in terms of the key geomorphic and roughness features was found by evaluating changes in random roughness at specific cross sections.

**Comment 1: Page 3, line 12: Maybe, you should give the geographical coordinates and elevation of your study area.**

Response:

The geographical coordinates and elevation of the experimental plot are now provided (Page 3, lines 27-29):

*"This study was conducted on an experimental plot (Fig. 1b) of the U.S. National Science Foundation Intensively Managed Landscapes Critical Zone Observatory in the headwaters of Clear Creek, IA (41.74º N, -91.94º W and an elevation of 250 m above mean sea level)."*

**Comment 2: Page 3, line 14: What do you mean by "mixed, superactive"?**

Response:

The terms "*mixed*" and "*superactive*" are soil taxonomy nomenclature used by the USDA (https://www.nrcs.usda.gov/Internet/FSE_DOCUMENTS/nrcs142p2_051232.pdf). The term "*mixed*" is related to the mixed nature of clay mineralogy, since these soils contain both smectite and illite. The term "*superactive*" is associated with the high degree of activity of cation exchange in the soil. The cation ion exchange capacity for these soils is between 15 and 30 Meq/100 g. Even though the goals of this paper do not include the effects of texture (since we only use one soil type) this information is helpful for the reader to judge the applicability of this study to their sites. We have added this info along with the textural characteristics of the soil (Pages 3-4, lines 29-1):

*"The soil series at the plot where the experiments were conducted is Tama (fine-silty, mixed, superactive, mesic Cumulic Endoaquoll) (http://criticalzone.org/iml/infrastructure/field-areas-iml/). It consists of 5% sand, 26% clay, 68% silt, and an organic matter content of 4.4%. The aggregate size distribution of the soil consists of 19% of the soil size fraction less than 250 μm, 48% between 250 μm and 2 mm, and 33% greater than 2 mm. These soils contain both smectite and illite, with high cation exchange capacity between 15 and 30 Meq/100 g."*

**Comment 3: Page 3, line 18: "to the plots", how many plots?**

Response:

There was one experimental plot. This was a typo, which we have corrected.

**Comment 4: Page 3, line 25: "widely accepted", by whom?**

Response:

The Marhsall-Palmer distribution is an accepted relationship by the American Meteorological Society (http://glossary.ametsoc.org/wiki/Marshall-palmer_relation). The following reference was added in the text:

Marshall, J.S., Palmer, W.M.K.: The distribution of raindrops with size, Journal of Meteorology, 5, 165–166, doi:10.1175/1520-0469(1948)005<0165:TDORWS>2.0.CO;2, 1948.

**Comment 5: Page 3, lines 27-29: This statement is too strong, even though rainfall characteristics were similar, other regions may have different soil types than the one studied here. What are the potential biases you are referring to in this sentence?**

Response:

We agree with the reviewer that the direct comparison of our results may only be possible under similar soils examined in this study. This includes regions exhibiting semi-humid climate conditions and have mollisol soils. We have added the following caveat to the Discussion and Conclusions (Page 10, lines 26-28) and the citation to the reference list:

"*It is recognized that dryer, silty type soils may not exhibit the increase in RR shown here. Also, the role of sealing may be important on roughness development under bare soil conditions and needs further examination. Soil water retention characteristics of the soils under sealing and its implication to RR must be considered (Saxton and Rawls, 2006).*"

In terms of other potential biases between natural rainfall and the rainfall supplied by our simulators, a proper calibration must be performed to match the drop size. Drop size affects the terminal velocity and the kinetic energy of the falling rain. A poor calibration of the raindrop size distribution would affect the overall size and shape of the roughness formed. The sentence (Page 4, lines 13-16) has been modified as follows:

"*This level of attention was taken to minimize any potential biases compared to natural rainfall with respect to raindrop size distribution, and, thus, render the rainfall simulation experiments scalable to other regions experiencing the same type of soil, bare surface, roughness conditions, and natural rainfall characteristics.*"

**Comment 6: Page 3, line 32: I would substitute "a priori the runs ensured that" for "before the runs confirmed that".**

Response:

This change was made.

**Comment 7: Page 4, line 1: This seems more like a result than materials and methods.**

Response:

We agree with the reviewer. This sentence was removed from the Materials and Methods.

**Comment 8: Page 4, line 3: What is CCD?**

Response:

It is a charge-couple device camera that serves to transfer the electrical charge to the attached computer and to convert it into digital signal. *"Charge-couple device"* was used in the text instead of *"CCD"*.

**Comment 9: Page 4, line 5: I would use "by software" instead of "from the desktop, using a computer program".**

Response:

The change was made.

**Comment 10: Page 4, line 7: Please, provide the names and references for the specific software used.**

Response:

The reference for the software is now provided in the text (Page 4, lines 25-28) as follows:

*"The information from each scan is converted into a set of (x,y,z) coordinates using a calibration file and the software developed from the USDA-ARS National Soil Erosion Research Laboratory for data transformation as explained by Darboux and Huang (2003)."*

**Comment 11: Page 4, line 12: Remove "experimental" before "tests".**

Response:

The word *"experimental"* has been removed.

**Comment 12: Page 4, lines 12-13: Change the sentence to "Rainfall intensities were respectively 30, 60 and 75 mm/h for experiments 1, 2 and 3".**

Response:

The text has been changed per the reviewer's suggestion.

**Comment 13: Page 4, line 14: Is the duration of your rainfall events the same of the storms you are referring here? In fact, 75 mm/h during 5 hours means 375 mm in 5 hours, which seems too much. What is the return period of these events? I mean, are these storms really so usual?**

Response:

No, the duration is not the same. However, because rainfall effects for the controlled experiments performed are isotropic it is the intensity that affects splash and RR (see all responses to comment 1 of reviewer 3). The 30 mm/h event has a recurrence interval of ~0.25-yr, while the 60 mm/h event has a 20-yr recurrence interval. Finally, the 75 mm/h event has a 60-yr return period.

Duration would only matter if the goal of the tests was to examine the role of runoff and shear on RR. Here the goal is to examine the role of KE on RR.

$$KE_i = \frac{1}{2} m_i v_{t\,i}^2$$

where $m_i$ is the mass of raindrop i (kg), $v_{ti}$ (m/s) raindrop terminal velocity, $\rho_i$ is the density of raindrop i (kg/m3), and $V_i$ is the volume of the raindrop i ($m^3$) which assumes a spherical shape.

During each of the experiments, the rainfall intensity remained constant and the storm duration was chosen in order to reach steady-state conditions in terms of infiltration and runoff and thus obtain repeatable results in terms of roughness patterns. Three replicates of each rainfall intensity were performed, and repeatability was confirmed by evaluation of changes in RR at specific cross-sections in the rainsplash dominated zone, as shown below for the 60 mm/h case. We found that on an average, the relative error of the RR ratios between replicates did not exceed 7%. Per the reviewer's comment, we have added (Pages 4-5, lines 34-3):

*"Three replicates of each rainfall intensity case were performed, and repeatability was confirmed by evaluation of changes in RR at specific cross-sections in the rainsplash dominated zone. It was found that on an average, the relative error of the RR ratios between replicates did not exceed 7%."*

|  | Pre-Rainfall RR (mm) | Post-Rainfall RR (mm) |
|---|---|---|
| 60 mm/h - Replicate 1 | 0.42 | 1.44 |
| 60 mm/h - Replicate 2 | 0.55 | 1.59 |
| 60 mm/h - Replicate 3 | 0.37 | 1.40 |

[Figure]

**Comment 14: Page 4, line 15: "Decagon soil moisture sensors", specify depth, how many and model.**

Response:

Six 5TE soil moisture sensors were placed to a depth of 10 mm along the whole plot (Page 5, lines 3-4):

*"The volumetric soil water content was recorded with via six 5TE soil moisture sensors manufactured by Decagon Devices, Inc. and placed along the whole plot to a depth of 10 mm."*

**Comment 15: Page 4, line 16: What do you mean by 35%? What is the field capacity of this soil? Please, specify.**

Response:

This value of 35% refers to the average initial volumetric water content at the plot. The field capacity for the soil is 38%. The sentence (Page 5, lines 4-6) reads as:

*"The initial volumetric water content was found to be the same for each experiment and approximately equal to 35% at the whole plot, where the field capacity of the specific soil is 38%."*

**Comment 16: Page 4, line 29: Remove "By definition".**

Response:

The phrase was removed.

**Comment 17: Page 4, line 31: "extracted" instead of "extract".**

Response:

The word was replaced.

**Comment 18: Page 4, line 32: "were" instead of "are".**

Response:

The word was replaced.

**Comment 19: Page 5, line 2: What is "its commonality found in the literature"?**

Response:

The sentence has been reworded to better convey our message (Page 5, lines 29-30):

*"The RR index calculated from Eq. (1) was used in this study as the principal method to quantify soil surface roughness due to its frequent and widespread use in various studies and landscape models as a descriptor of microroughness."*

Please note that additional indices were considered per the request of Reviewer 2.

**Comment 20: Page 5, line 3: "was used" instead of "is used".**

Response:

The phrase has been replaced.

**Comment 21: Page 5, lines 8-12: This is not clear to me. Did you use all these methods? At the end of the sentence you say "among others", what do you mean? Do you imply that you used more methods than those indicated here?**

Response:

We see how confusion may have arisen from our original statement.  No, we did not use all the methods outlined. We have modified the sentence to read (Pages 5-6, lines 31-2):

*"If correlation exists within a certain spatial scale, RR will likely change with the changing window size of observed data (Paz-Ferreiro et al., 2008) and may be dependent on the resolution of the measurement device (Huang and Bradford, 1992).  Thus, alternative scale-independent methods that consider spatial correlation have been developed by other researchers in order to address this issue."*

**Comment 22: Page 5, line 14: I would split this sentence in two. Instead of "with the advantage of its quantification being scale independent", I would use a point and then "It has the advantage of being scale independent".**

Response:

The sentence was modified as suggested.

**Comment 23: Page 5, line 16: "the semivariogram is a useful…" I do not think this sentence is really needed.**

Response:

The sentence was removed.

**Comment 24: Page 5, lines 22-25: it seems rather peculiar that you explain what a semivariogram is but not what the Hurst exponent indicates.**

Response:

The Hurst exponent is briefly explained after Eq. (3). However, it is not extensively described because it is not of particular interest for this study. It has been shown to be less sensitive than crossover length when describing soil surface evolution influenced by rainfall (Vázquez et al., 2005). The text now reads as (Page 6, lines 23-24):

*"The generalized Hurst exponent is a less sensitive descriptor of soil surface evolution as influenced by rainfall (Vázquez et al., 2005), hence attention is mostly centered on the crossover length."*

**Comment 25: Page 5, line 28: "and 0 < H < 1", this should come before, when you refer to H and not after the crossover length.**

Response:

This changed has been made.

**Comment 26: Page 5, lines 32-33: I did not understand this sentence. Please, re-phrase it.**

Response:

We rephrased these two sentences (Page 6, lines 24-27), which now read as follows:

"*Given the semivariogram plot calculated using Eq. (2), H and l can be extracted by fitting a power law relationship in the form of $y = Ax^B$ to the semivariance-lag distance data, where $y = \gamma(h)$ and $x = h$. According to Eq. (3), the B regression variable gives the generalized Hurst exponent value and the A regression variable yields the crossover length.*"

**Comment 27: Page 6, lines 1-4: Could you, please, re-phrase this paragraph? I do not understand it properly, it is a bit confusing.**

Response:

We have completely removed this paragraph to avoid confusion of the reader. Besides, it is explained in the Results section, please see Comment 8.

**3. Results**

**Comment 1: Page 6, lines 6-14: This looks more like materials and methods than results.**

Response:

We moved and modified this paragraph to the end of the section of Materials and Methods (Page 7, lines 12-18):

"*In order to negate the effects of the differences found in initial microrelief amongst the three runs and compare rainfall induced changes in relative terms, the results from the rainfall experiments are presented in the form of ratios of the roughness indices. More precisely, the RR ratio, defined as the ratio of the RR index post-rainfall over the RR index prior to the rainfall ($RR_{post}/RR_{pre}$), is calculated for each experiment. Semivariograms are plotted under pre- and post-rainfall conditions at the ROI to assess the spatial correlation of surface elevations. Along the same lines, ratios between pre- and post-rainfall conditions are calculated for the crossover*"

*length, the variance length scale of the Markov-Gaussian model, and the limiting difference to assess changes in microroughness along with the RR ratio."*

**Comment 2: Page 6, lines 12-14: Have these comparisons been performed against literature data? Did this literature provide ratios?**

Response:

Yes, sections 3.1 and 3.2 are now updated and include more detailed comparisons with the cited studies. Figure 5 is updated to show the RR ratio versus the initial value of RR for our study along with the relevant studies considered. From Fig. 5 it is now clear that our study captures the behavior of RR for an initial range that was not covered before. In some of the studies (e.g., Zhang et al., 2014), ratios of the roughness indices were already provided, whereas in others they were calculated, since only the initial and final values of the indices were provided. Moreover, the study of Vázquez et al. (2008) which was previously missing from our study is now added to Fig. 5 and Table 1 and discussed in the text for completeness, since it involves quantification of RR for nearly smooth surfaces. Finally, cumulative rainfall amounts for each experiment are provided in Table 1 for a better inter-study comparison.

The updated Results section for the inter-study comparisons now reads as (Pages 7-8, lines 23-2):

*"Figure 5 shows the RR ratio, i.e., $RR_{post}/RR_{pre}$, with respect to the initial value of RR for the present study along with other studies that quantify rainfall induced microroughness changes. The dashed line at the RR ratio value of unity reflects no change in roughness, thus all points above that line show an increasing trend with rainfall, while all points below show a decreasing trend with rainfall. All the studies capture a wide range of initial RR values – up to 21 mm – and it is clear that our study captures the behavior of RR for an initial range that was not covered before. Figure 5 suggests that roughness may increase with raindrop impact for a range of low initial RR values (< 5 mm), while it consistently decays for high initial RR values (> 5 mm). It is acknowledged that the values of the roughness indices among different studies may involve some experimental error and may reflect different conditions (such as rainfall forcing and soil type). Specifically, Vázquez et al. (2007) used clay textured soil, Vázquez et al. (2008) used silt loam textured soil, while our study along with all the other studies cited conducted rainfall experiments for silty clay loam textured soil. Rainfall intensities and cumulative rainfall amounts varied significantly among studies."*

The revised Figure 5 is below:

[Figure]

**Figure 5: Random Roughness (RR) Ratio versus initial RR for this study and other selected studies.**

**Comment 3: Page 6, line 20: Have these experiments been performed only once?**

Response:

This question was answered earlier under comment 13 in the Materials and Methods. Three replicates of each rainfall intensity case were performed.

**Comment 4: Page 6, lines 21-23: Please, re-phrase this. It seems redundant.**

Response:

The whole paragraph (Page 8, lines 3-19), which contained this phrase was modified to put our study into a context with the available literature. In doing so we eliminated the part that was considered redundant.

"*Table 1 summarizes the results of this study along with results from the selected studies in quantitative terms, documenting the RR index values before and after the rainfall events, the cumulative rainfall, as well as the associated RR ratio. Two inferences can be made from Table 1. First, our study along with Vázquez et al. (2008) and Zheng et al. (2014), which were*

*performed for the smooth surface initial condition, report an increase in RR with rainfall for the most part. Exception seems to hold for one soil surface of the study of Vázquez et al. (2008), as well as the smooth surfaces of Vermang et al. (2013) which show decaying roughness due to rainfall. Second, the present study indicates that the RR ratio becomes higher with higher cumulative rainfall amount when the surface is classified as smooth, whereas the opposite tends to hold for soil surfaces classified as disturbed (Fig. 5, Table 1). Vázquez et al. (2008) and Zheng et al. (2014) recorded an increase in RR with rainfall but had significantly lower values of RR ratio than we did. This may be attributed to the fact that they applied lower cumulative rainfall amount, and the initial microroughness conditions in their study were higher. Other studies that are not included in Table 1 have also shown increasing trends of roughness with rainfall, as quantified by the use of different indices. For instance, Huang and Bradford (1992) calculated the semivariograms for different surfaces and used fractal and Markov-Gaussian parameters to quantify the roughness. Markov-Gaussian analysis showed a relative increase in the roughness parameter for a surface of low initial roughness. Finally, Rosa et al. (2012) introduced the Roughness Index which is estimated from the semivariogram sill in order to quantify roughness, and observed its increase with rainfall under low initial roughness conditions. That increase was attributed to the fragmentation of aggregates and clods to smaller aggregates but was not linked to smooth bare soil surface conditions."*

**Comment 5: Page 7, lines 4-5: How did you check this significance? Only by stating that difference is less than 10%?**

Response:

A nonparametric test for spatial isotropy was performed per Guan et al., 2004 using the public domain R statistical package with 'spTest' library. A p-value less than 0.05 for all cases confirmed the spatial isotropy hypothesis. Thus, there would be no bias by taking one representative direction to calculate the semivariograms and the associated crossover lengths. The corresponding part of the paragraph was adjusted as follows (Page 8, lines 26-30):

*"Since the action of rainfall is isotropic and adds no systematic trend along any direction, no significant differences were expected between semivariograms. A nonparametric test for spatial isotropy was performed per Guan et al., 2004 using the public domain R statistical package with the 'spTest' library. The spatial isotropy hypothesis was confirmed ($p < 0.05$). Thus, there would be no bias in taking any direction to calculate the semivariograms and the associated crossover lengths."*

**Comment 6: Page 7, line 6: "representative semivariogram", that corresponds to which angle?**

Response:

The semivariograms were chosen at the downstream direction that corresponds to an angle of 0 degrees. We have removed the term "representative".

**Comment 7: Page 7, line 7: "semivariograms" instead of "semivariogram".**

Response:

This change was made.

**Comment 8: Page 7, lines 10-11: "which is considered sufficient to assume no spatial autocorrelation at the scale examined in this study", I do not get this; you checked all roughness for the 200 mm, so you should have accounted for lag distances less than 10 mm.**

Response:

The semivariograms describe the mean variance of the elevation measurements for a range of lag distances and indicate the distance at which spatial autocorrelation ceases to exist.  RR is independent of the window size, only if the window size significantly exceeds the spatial autocorrelation range, which is the case here. Our intent here was to point out that the window size of the ROI is sufficiently large enough to eliminate the effects of spatial autocorrelation of data on RR quantification. The last part of the paragraph (Pages 8-9, lines 33-5) has been rephrased as follows:

*"These lag distances are approximately 10 mm, so the selected 200 mm window size of the ROI is almost 20 times greater than the spatial autocorrelation range.  This implies that the window size of the ROI falls at the scale of the "semivariogram sill" (which is defined as the near-constant value of semivariance at large lag distances where the semivariogram levels out – see horizontal dashed lines in Fig. 6).  RR is directly related to the semivariogram sill (e.g., Vázquez et al., 2005; Vermang et al., 2013), therefore it is independent of the selected window size, given that the latter significantly exceeds the spatial autocorrelation range."*

**Comment 9: Page 7, lines 16-17: "pre-rainfall values" instead of "pre-rainfall value for all three intensities".**

Response:

The change was made.

**Comment 10: Page 7, line 18: I would use "events" instead of "precipitation intensities".**

Response:

The change was made.

**Comment 11: Page 7, line 19-20: Please, check English and re-phrase this sentence.**

Response:

The (Page 9, lines 9-11) sentence has been reworded as follows:

"*Complete agreement between the trends of the RR index and the semivariogram sill justify the use of the RR index as a representative and unbiased descriptor of microroughness.*"

**Comment 12: Page 7, line 22: Remove "in existing literature".**

Response:

The phrase has been removed.

**Comment 13: Page 7, line 24: "reported" instead of "report".**

Response:

The change was made.

**Comment 14: Page 7, line 25: "found" instead of "in the crossover length reported".**

Response:

This has been done.

**4. Conclusions and Discussion: This section is very weak and should be greatly improved. Besides, it should be entitled "Discussion and Conclusions".**

Response:

The title was changed to "Discussion and Conclusions". A great effort was made to improve this section in response to the reviewer's valuable comments. Please see the specific responses below.

**Comment 1: Page 8, line 2: Are you sure that these experiments are "unique and novel". I am also concerned about the fact that you state that your experiments "mimic natural rainfall conditions" but you never described those natural rainfall conditions.**

Response:

Very few studies have been designed to assess microscale variations under controlled conditions and thus record increases in RR with rainfall intensity. There are several rainfall simulator experiments out there; however, our experiments are unique in the sense that they were designed to help us decipher the role of rainsplash on RR by isolating it from the role of other processes such as runoff, variable water content, bare soil surface, texture, etc. They also mimic natural rainfall conditions, as described in the Material and Methods (Comment 4). We have modified the text to better describe the uniqueness of our experiments (Page 10, lines 2-5):

"*Very few studies have been designed to assess microscale variation under controlled conditions to purposely examine increase in RR with rainfall intensity. Unique experiments are presented herein that were designed to help us decipher the role of rainsplash on increasing RR by isolating the role of other processes such as runoff, variable water content, bare soil surface, soil texture, etc.*"

**Comment 2: Page 8, lines 5-6: "which are confirmed as reliable descriptors of microroughness"; this is already known.**

Response:

This phrase was removed.

**Comment 3: Page 8, lines 7-9: I have doubts about this, you only performed your experiment once and considered a small surface where raindrop detachment prevails over runoff; were the same conditions in the other studies? Did they consider only raindrop detachment?**

Response:

Experiments were performed 3 times. But we agree with the gist of your comment.

These lines probably come across too strong and may not reflect the main message of this study. As the spatial scale increases the effects of rainfall may not be dominant with time. When considering larger areas, runoff and concentrated flow (i.e., rills) will become more prevalent.

However, we all have to agree that during a storm, especially in the early part, there is a period when rainfall action may be more important than runoff, independent of location. It is hard to pinpoint the duration of that period because in some instances the soil may be saturated and runoff dominates right away. During that initial period, though, modeling the evolution of RR is important. This period is the focus of our controlled experiments and this study. The upslope area of the experimental plot provides the controlled conditions. We have therefore taken extra steps following a few other available studies (e.g., Zheng et al. 2014) to design tests where we can isolate the effects of rainfall on RR evolution. We discovered that RR does not always decay with intensity (and the kinetic energy of the raindrop) at all times but depending on the initial surface micorroughness condition RR can increase with time. This process can lead to the

formation of depression storage which can ultimately affect ponding and lead to the formation of flow pathways.

All studies that reported on microroughness were performed at similar size experimental plots because this way they can isolate the effects of raindrop from runoff. Small plots suggest less area for runoff and formation of concentrated flow pathways. This is the reason in fact we have focused on a smaller section within the upper part of the plot where RR effects were dominant.

We have rephrased this statement to avoid confusion as follows (Pages 10, lines 5-10):

*"The results obtained are consistent with findings of other studies that have examined length scales up to 5 mm. These length scales (i.e., ~2-5 mm) have been found to be common in agricultural landscapes due to prolonged exposure to rainfall impact, runoff and freeze-thaw cycles. Within these landscapes, the reported increase is expected to occur during the early part of the storm where rainsplash action may be more important than runoff."*

**Comment 4: Page 8, lines 11-12: What are the implications of this?**

Response:

Please see the response to Comment 5 below, where the implications of localized microroughness residuals are better highlighted.

**Comment 5: Page 8, line 13: "Roughness residuals infer depression storage residuals", what do you mean?**

Response:

We suggest that localized microroughness residuals as shown from this study will leave additional depression storage at the surface prior to runoff generation. This may alter ponding and flow pathway patterns for storm events. The sentence (Page 10, lines 14-16) was altered to the following and the citations were added to the reference list:

*"Increase in microroughness further infers increase in depression storage at the soil surface prior to runoff generation (Kamphorst et al., 2000), which can significantly alter the ponding and flow pathway patterns especially at the onset of a storm event (Onstad, 1984)."*

**Comment 6: Page 8, lines 15-20: I am not sure about this. Your results come from a limited number of experiments and you are implying that they have a strong importance in various disciplines and applications… it seems overestimating your findings.**

Response:

We understand the reviewer's hesitation here and the potential overestimation of the findings' importance. This study's results, although being limited due to the level of work needed to

complete them, consistently report a similar finding, which has been essentially undocumented to date. We feel that it will benefit and possibly further motivate environmental modeling and research, although maybe not directly other disciplines and applications. This part (Page 10, lines 16-18) has been modified as follows:

"*Our findings provide a better understanding of the highly dynamic phenomenon of soil surface microroughness evolution under the impact of rainfall. Our study motivates further research on the extent of influence of the examined phenomenon and its mathematical formulation for modeling applications.*"

**Comment 7: Page 8, lines 21-23: This must be further explained, I do not see your point.**

Response:

We point out that soil surface roughness both depends on and affects hydrologic response. Localized increases in microroughness create additional depression storage at the surface (Kamphorst et al., 2000) with potential implications to flow and pathway patterns (Gómez and Nearing, 2005). However, the extent to which soil surface roughness increases would affect depression storage, runoff, and flow pathways is unknown and further research to quantify this effect is needed. To be more concise, the last part of the paragraph was modified accordingly (Pages 10-11, lines 31-4) and the citations added to the reference list:

"*Finally, this study and other studies demonstrate that the evolution of soil surface roughness in response to rainfall is dependent on initial roughness conditions and rainfall intensity and can contribute to hydrology, i.e., another factor shaping the soil surface (e.g., through runoff). Different behavior of surface roughness evolution, i.e., increase or decrease, depending on initial roughness conditions indicates a dynamic and nonlinear feedback between hydrologic response and surface roughness which may affect depression storage, ponding and flow pathways (Kamphorst et al., 2000; Gómez and Nearing, 2005). However, the extent to which soil surface roughness increase would affect depression storage, runoff, and flow pathways is unknown and further research to quantify this effect is needed.*"

**Comment 8: Page 8, line 24: Remove "study's".**

Response:
It has been removed.

**Comment 9: Page 8, lines 23-25: I really think that you are overestimating your results.**

Response:
It has been shown through our own research and others (e.g., Gómez and Nearing, 2005) that due to the demonstrated interplay between roughness and hydrology at the microscale, we will see

changes in runoff in terms of magnitude and timing. Most physically based models (e.g., WEPP) which assume a decrease in roughness following a rain event in all cases will have an error in the estimations of runoff and hence erosion. We do not infer that lack of consideration of our findings will necessarily or drastically affect hydrologic response. However, the addition of our results may address some of the discrepancies in the results. The last sentence (Page 11, lines 4-5) has been added:

"*Finally, this study and other studies demonstrate that the evolution of soil surface roughness in response to rainfall is dependent on initial roughness conditions and rainfall intensity and can contribute to hydrology, i.e., another factor shaping the soil surface (e.g., through runoff). Different behavior of surface roughness evolution, i.e., increase or decrease, depending on initial roughness conditions indicates a dynamic and nonlinear feedback between hydrologic response and surface roughness which may affect ponding and flow pathways (Gómez and Nearing, 2005). However, the extent to which soil surface roughness increase would affect depression storage, runoff, and flow pathways is unknown, and further research to quantify this effect is needed. Nonetheless, the current findings may help explain some modeling discrepancies in terms of depression storage and runoff predictions.*"

**Comment 10: Page 8, lines 27-29: Yes, alright but is the initial roughness less than 2 mm? Besides, you indicate that Intensive Managed Landscapes have bare soil 75% of the time; it looks not very intensive…**

Response:

As noted above, increase in surface roughness has been recorded for surfaces with initial microroughness of the order of 2-5 mm. Several studies have shown that landscapes with surface roughness of this order of magnitude are common in agricultural landscapes (Burwell et al., 1963; Allmaras et al., 1966; Burwell et al., 1969; Cogo, 1981; Currence and Lovely, 1970; Steichen, 1984; Unger, 1984; Zobeck and Onstad, 1987) due to long exposure to raindrop impact, runoff and freeze-thaw cycles (Zobeck and Onstad, 1987; Abaci and Papanicolaou, 2009).

Depending on the management (i.e., tillage intensity), the period between harvest and planting where the surface cover is bare will vary from 40 days to 6 months. It is understandable, when seeing the long period of inactivity between harvest and planting, to have misgivings with the term "intensively managed landscapes". However, even though intensively managed landscapes have essentially bare soil conditions from harvest to planting, the level of work that goes into planting, harvest, fertilization and other amendments is quite extensive in a short window. This is also quite demanding for the soil and microbes living in the soil, thus the term "intensive".

**Comment 11: Page 8, line 33: "landscape response to precipitation"; however, your study refers only to 200 mm$^2$ surface… is this not overestimating your results?**

Response:

We understand the reviewer's concern. We were lax with our wording and speaking generally. The term "landscape response" may be a stretch, especially since our paper deals heavily with scales. As mentioned before, we believe that our findings are applicable to the early parts of storm events when rainsplash effects are most dominant for soil surfaces with roughness on the order of 2-5 mm. The sentence (Page 11, lines 12-14) has been reworded as follows:

*"To the extent that microscale processes are considered significant, we argue that such models should adequately capture the increasing and decreasing trends in soil microroughness during all stages of a storm event in order to accurately predict local response to precipitation."*

**Comment 12: Page 9, line 1: Again overestimating the importance of your results. How this slight increase in RR may affect erosion processes?**

Response:

Indeed, the extent to which the increase in RR recorded herein can affect erosion processes is not yet known. However, it has been noted that different values of RR can affect flow pathways and runoff, which consequently can affect erosion processes (Gómez and Nearing, 2005). The above has been documented in the text (Page 11, Lines 14-16):

Furthermore, the following sentence has been removed in order not to overstate the importance of our results of the study:

""

**Comment 13: Page 9, line 4: This needs, at least, a reference.**

Response:

This section was deleted in order to maintain better focus on the main message of the study.

**Comment 14: Page 9, line 5: "new statistical analyses", what statistical analyses did you perform?**

Response:

Again, in order not to divert from the main message of the study, we have removed the sentence.

**Comment 15: Page 9, line 6: I am not sure about what you mean by "is present a priori".**

Response:

The sentence was modified as (Page 11, lines 16-17):

"*In fact, the majority of existing models assume that RR always decays over time with rainfall.*"

**Comment 16: Page 9, line 7: "in the current paper" instead of "in the paper".**

Response:

The change has been made.

**Comment 17: Page 9, line 9: "is improved for current models", do you mean that is already done in current models?**

Response:

Many existing models have parameterizations for the evolution of surface roughness (e.g., the Water Erosion Prediction Project, WEPP; Flanagan and Nearing, 1995). However, they do not account for the possibility that under certain microroughness conditions, rainfall may increase in roughness. Therefore, our study is calling attention to the need that current and future models must account for this condition of increasing roughness with rainfall over a smoothened surface. The ratios provided in this study are good first step for improving these models. The sentence (Page 11, lines 18-20) was reworded as:

"*By providing the ratios of increase in roughness indices with rainfall intensity, the parameterization of the evolution of surface roughness with rainfall could be improved for current models.*"

Flanagan, D. C. and M. A. Nearing. USDA-Water Erosion Prediction Project: Hillslope profile and watershed model documentation, 10, NSERL report, 1995.

**Comment 18: Page 9, lines 11-12: "extension of the experiments in areas such as downslopes where concentrated flow and rilling are of importance" That you did not want to account in your study although you could have done in view of the surface of your experimental plot.**

Response:

In this study, we examined only a small slice of the problem by isolating the effects of rainfall on roughness for smooth bare soil surfaces to get a better understanding on this phenomenon. It was considered as an essential first step before extending the study to include the combined effects of flow concentration, rilling, and rainfall on surface roughness. The last sentence (Page 11, lines 20-23) has now been improved based on the above:

*"Future research will complement the present efforts by quantifying the evolution of microroughness under the collective action of rainfall and runoff. This will require a good understanding and quantification of the extent to which the initial increase in roughness in the early part of the storm could have an impact on flow pathways, runoff, and processes at subsequent parts of the storm."*

**5. References**

**Comments:**

**Eight references are not cited in the text. Please, check them and also edit the reference list according to the journal guidelines.**

**Page 10, lines 17-18: Chu et al. (2012) is not cited in the text.**

**Page 10, lines 20-23: Why did you use upper-case letters for the title of these publications?**

**Page 10, line 27: Why did you use capital letters for CATENA?**

**Page 10, lines 30-31: Why did you use upper-case letters for the title of this publication?**

**Page 11, lines 1-3: Le Bissonais (2016) is not cited in the text.**

**Page 11, lines 4-5, 10 and 17-18: Why did you use upper-case letters for the title of these publications?**

**Page 11, line 23: Why did you use capital letters for SOIL ORGANIC CARBON DYNAMICS?**

**Page 11, line 28: Potter (1990), this reference does not follow the style of the journal; the year of publication should come at the end.**

**Page 11, line 29: Why did you use upper-case letters for the title of this publication?**

**Page 11, line 32: Why did you use capital letters for CATENA? Besides, Römkens et al. (2002) is not cited in the text.**

**Page 12, lines 4-5: Remove the quotation marks.**

**Page 12, lines 8-9: Vázquez et al. (2006) is not cited in the text.**

**Page 12, line 11: Remove "European Geo-sciences Union (EGU)".**

**Page 12, lines 15-17: Vázquez et al. (2010) is not cited in the text.**

**Page 12, lines 18-19 and 25-28: Why did you use upper-case letters for the title of these publications?**

**Page 12, lines 22-28: Zhao et al. (2014), Zhao et al. (2016) and Zheng et al. (2012) are not cited in the text.**

Response:

For the comments regarding the references, we have addressed them all as requested by the reviewer. In summary, we consistently followed the journal guidelines in terms of formatting, removed the uncited references, and added all new references during the major revision process of the manuscript. Please see the updated version of the References (Pages 12-15).

**6. Figures**

**Comments:**

**Figure 1: Modify the caption to "(a) Types of soil surface microroughness. (b) Experimental plot. The rainfall simulator is placed above the bare soil surface and a base made of wood is put into place to facilitate the movement of the surface-profile laser scanner".**

**Figure 2: Modify the caption to "Setup of the experimental tests: (a) Rainfall simulators are mounted in series and a pump provides them with water from a tank; (b) rainfall simulators are placed and adjusted at a height of 2.5 m above the experimental plot surface to ensure drop terminal velocity is reached".**

**Figure 3: Indicate in the caption the interpolation technique that was used.**

**Figure 4: Why not a and b as in the former figures and you used left and right? You should define ROI in the caption. Besides, why "part"? If the whole experimental plot was 7 x 1.2 m is the whole plot what you are representing in the right-hand side of the figure and not only part of it.**

**Figure 5: Remove "considered herein". I think that you do not need to include experiments 1, 2 and 3 if you indicate the rainfall intensity. Remove the border of the figure and the second decimal from the Y-axis.**

**Figure 6: Remove "Spatial" and "considered herein". Use "region of interest" instead of "ROI".**

**Figure 7: Remove "considered herein". I think that you do not need to include experiments 1, 2 and 3 if you indicate the rainfall intensity. Remove the border of the figure and the second decimal from the Y-axis.**

Response:

Regarding the comments to the figures, which are listed above, we have addressed them all as requested by the reviewers. These comments were all essential, but for the most part cosmetic and do not need further elaboration. Figure 5 was updated and is described in the Results Comment 2 above. Finally, Figure 7 was removed since it was considered as unnecessary. Please see the updated version of the figures (Pages 16-20).

---

## Author Comment (AC2) · 19 Mar 2017

**RESPONSE TO REVIEW COMMENTS**

We thank the reviewer for the valuable input, which has helped improve the quality of our manuscript.    Our responses are provided below.  **Please note that the original comments are in black letters and our responses are in blue letters**.  In addition to these responses, we will provide a revised manuscript that reflect the proposed changes, as well as a copy with the tracked changes where revisions were implemented.

In summary:
1. We have significantly enhanced the flow, clarity, and precision of the text. The abstract is also very clear in terms of objectives, methodology, and findings.
2. We have put all our results into context, by providing all the relevant literature that has quantified soil surface roughness at the examined bare smooth soil surface conditions, explicitly acknowledging the studies with results that show the increase in roughness and added missing references (e.g., Kamphorst et al., 2000; Vázquez et al., 2008). We have also updated Fig. 5 to depict the changes seen in RR with respect to initial RR from our study and other studies.
3. We discussed the advantages by focusing on a single rainfall event rather than successive events in the context of this study.
4. We have added information regarding the soil characteristics considered in the study.
5. We have provided two additional commonly used indices for soil surface roughness. Their values and trends with rainfall are in good agreement with RR and crossover length, and in support of our conclusions.
* * *
*General Comments:*
**This study analyses soil surface roughness evolution after a single event of simulated rainfall with three different intensities, namely 30, 60 and 75 mm/h. Two indices used to describe the magnitude of soil surface roughness indicate increasing values of this variable after rainfall addition.**

**In my opinion this manuscript does not contain significant results. This is because the experimental work has been limited to one rainfall event, and this is obviously a main weakness in any study about soil surface roughness evolution (either increase or decay). In addition, authors claim that the results are new, as they state that i) "Findings show a consistent increase in roughness under the action of rainfall for initial microroughness length scales of 2 mm" and ii) "This contradicts existing literature where a monotonic decay of roughness of soil surfaces with rainfall is recorded for disturbed surfaces". However, please note that i) again, the increase in roughness (instead of a expected decrease) has been found only for the first event. What about successive events); results are not reported. ii) Increases in soil surface roughness after simulated surface rainfall and for**

disturbed soil surfaces have been previously reported (Please, see Vidal Vázquez et al., 2008. Assessing soil surface roughness decay during simulated rainfall by multifractal analysis. Nonlin. Processes Geophys., 15, 457–468). In this paper the evolution of the surface of three different soils was studied during successive events; two of the studied soils showed soil surface roughness increased after the first event (similar to your results)e second, but it decreased after the second and successive events; the third soil studied showed scarce trend to either increasing or decreasing surface roughness values following successive rainfall events.

*Response:*

There are a number of different reasons why our study focuses on the evolution of soil surface roughness under single storm events. First, the goal of our experiments is to offer generic, controlled conditions to isolate the effects of raindrop impact on roughness from other processes (i.e., runoff). We meet this goal by checking how raindrop impacts roughness under representative for the region rainfall intensities. We are focusing on single events as these experiments allow us to control the antecedent soil moisture conditions and initial bed surface structure. All single storm event experiments start from the same antecedent soil moisture conditions and initial bed surface structure to facilitate comparisons under different intensities and enable comparisons with the reported literature. We acknowledge that a single storm event represents a rather idealistic case scenario and that a series of storm events can be important and should be examined in future studies.

**Specific Comments:**

**Comment 1: The obtained results should be put into context, with relevant references. This opinion is based in the fact that relevant studies about soil surface roughness, (including the previously cited Vidal Vázquez et al., 2008. Nonlin. Processes Geophys., 15, 457–468, and Kamphorst et al. 2000. Soil Science Society of America Journal 64(5): 1479.1458. By the way, these two manuscript present examples of soil surface roughness assessed by laser scanner as in your work. In the first work quoted (Vidal Vazquez et al., 2008) the magnitude of the roughness is not very different from that in your work and in the second (Kamphorst et al., 2000) several plots also are representative for conditions of rather low values of roughness.**

*Response:*

Per the reviewer's suggestion, the results of our study have been put into context, adding the relevant references related to rainfall simulation experiments and the associated evolution of soil surface roughness for smooth surfaces.

First of all, we expanded our results and discussion to elaborate on the comparison with other studies. The study of Vazquez et al. (2008) has been added along with the already cited work of Huang and Bradford (1992), Rosa et al. (2012), and Zheng et al. (2014), all of them providing indications that under certain conditions, roughness may increase with rainfall (See Page 8, lines 3-19 and Table 1).

We have created Figure 5 to better reflect the relevance of our results. It has been updated to show the rainfall induced relative change in RR with respect to the initial RR of the soil surfaces from our study and the studies outlined above. It is suggested that roughness may increase with raindrop impact for a range of low initial RR values (< 5 mm), while it consistently decays for high initial RR values (> 5 mm). It is also clear that our study captures the behavior of RR for an initial range that was not covered before.

[Figure]

**Figure 5: Random Roughness (RR) Ratio versus initial RR for this study and other selected studies.**

Findings of Vázquez et al. (2008) have been added to Table 1. In addition, our results are discussed in the light of the other cited studies in the Results section (Page, lines):

Finally, the study of Kamphorst et al. (2000) has been added to the Discussion and Conclusions section of the manuscript in support of the relationship between microroughness and depression storage (Page 10, lines 14-16):

*"Increase in microroughness further infers increase in depression storage at the soil surface prior to runoff generation (Kamphorst et al., 2000), which can affect ponding and flow pathway patterns especially at the onset of a storm event (Onstad, 1984)."*

**Comment 2: In my opinion, adding more experimental data (successive events) would allow that this manuscript reaches international standards.**

*Response:*
We do acknowledge the importance of accounting for successive events but this is not the focus of the specific study as stated earlier. We are therefore not dismissive of the reviewer's request.

Having said that, it is also important to note that the labor and level of detail required to perform the experimental runs presented in the study is significant, as it can be seen in Fig. 1b and 2. It takes about 10 days to prepare and run each test. This is done for 9 runs, so roughly a period of 3 months.

**In addition, any revision of this manuscript should address the following points:**

**Comment 3: Text should be ameliorated the text, which is not precise and provide a more clear presentation.**

*Response:*
The text was substantially improved in terms of clarity, grammar, language and structure. A significant amount of effort has been put to enhance the flow and precision of the text. A number of modifications include: Correction of grammatical errors. We moved and modified the first paragraph of the Results of the previous version of the manuscript to the Materials and Methods section, since it is more relevant there (Page 7, lines 12-18). The last section was more appropriately renamed to Discussion and Conclusions. Results are presented in a clear manner, as explained in Comment 1 above.

**Comment 4: Main corrections are expected in abstract, objectives and discussion and conclusion sections.**

*Response:*

The abstract was significantly improved to clearly present our objectives, methodology, and findings (Page 1, lines 13-28). It is specified that our study focuses on a bare smooth soil surface in an agricultural field. Our study is also put into context with existing literature, and the need to consider the cases where roughness can increase is highlighted, in light of the scarcity of studies that explicitly deal with rainfall induced change in roughness for the examined microroughness scales. Results from the additional indices examined were also added (see Comment 8).

The objectives of our study are now clearly stated (Page 3, lines 20-24):
*"The key specific objectives of this study are (i) to quantify the soil surface microroughness of smooth bare soil surfaces before and after the effect of rainfall, and (ii) calculate the relative change in roughness for different intensities. To meet the two specific objectives we employ four commonly used indices, the RR index, the crossover length, the variance scale from the Markov-Gaussian model, and the limiting difference. The last three indices are alternate methods and used here to supplement the RR index analysis for relative change in roughness."*

**Major revision has been made in the Discussion and Conclusion sections** (see Responses to Reviewer 1). In a nutshell, comments where it seemed that we stretched too far in the relevance of our results have been modified or removed, to be more in line with the level of the analysis. Furthermore, we provide a better insight into the significance of our study, stating that our experiments were designed to isolate the role of rainsplash on roughness from other processes such as runoff, variable water content, bare soil surface, texture, etc. Through our study we were able to determine that microroughness and its change are significant when there is no cover, which tends to happen between harvest and planting, and at the beginning of a storm event. We also provide in a clearer manner the limitations of our study, as well as the next steps for further research in terms of a better understanding and quantification of the extent to which the initial increase in roughness in the early part of the storm could have an impact on flow pathways, runoff, and processes at subsequent parts of the storm.

**Comment 5: Mechanisms and reason for the increase in soil surface roughness after one event simulated rainfall.**

*Response:*
Changes in roughness during a storm event can be attributed to compression and drag force from the raindrop impact on the soil, angular displacement due to rainsplash, aggregate fragmentation, and differential swelling (Al-Durrah and Bradford, 1982; Warrington et al., 2009; Rosa et al., 2012; Fu et al., 2016). To the best of our knowledge, no study has quantified the co-play of the outlined processes as influenced by different soil types, rainfall characteristics (e.g., median diameter of raindrop), and initial roughness conditions. Therefore, the exact mechanisms and reasons that lead to the increase in soil surface roughness are unknown.

We now acknowledge the above in the manuscript (Page 10, lines 23-26):

*"The exact mechanisms leading to increase in roughness are unknown. Changes in roughness during a storm event can be attributed to compression and drag force from the raindrop impact on the soil, angular displacement due to rainsplash, aggregate fragmentation, and differential swelling (Al-Durrah and Bradford, 1982; Warrington et al., 2009; Rosa et al., 2012; Fu et al., 2016)."*

**Comment 6: There are also unnecessary figures, regarding the experimental setup, as the methodology employed has been largely described before.**

*Response:*

Thank you for the comment. Other reviewers have requested that we put more information about the experimental set-up. We deem that our figures regarding the experimental setup provide the reader with the necessary information and specifics to ensure repeatability of the experiments outlined. Future research may require the repetition of the same experiments to study the coevolution and interaction between rainsplash and runoff, in order to further determine their collective influence on the hydrologic processes. We have adjusted other sections of the paper if the concern relates to space. Specifically, we have removed Figure 5 of the previous version of the manuscript and added the Figure described in Comment 1.

Finally, we have removed Figure 7 since it was considered unnecessary.

**Comment 7: Soil composition and main characteristics should be also reported in the material and methods section.**

*Response:*

We have added more information on the soil used in our study (Pages 3-4, lines 29-1):

*"The soil series at the plot where the experiments were conducted is Tama (fine-silty, mixed, superactive, mesic Cumulic Endoaquoll) (http://criticalzone.org/iml/infrastructure/field-areas-iml/). It consists of 5% sand, 26% clay, 68% silt, and an organic matter content of 4.4%. The aggregate size distribution of the soil consists of 19% of the soil size fraction less than 250 μm, 48% between 250 μm and 2 mm, and 33% greater than 2 mm. These soils contain both smectite and illite, with high cation exchange capacity between 15 and 30 Meq/100 g."*

**Comment 8: Other significant roughness indices should be addressed, in addition to random roughness and crossover length.**

*Response:*

Per the reviewer's suggestion, we examined additional indices other than the RR and crossover length, which can capture soil surface roughness at the examined scales. Specifically, the variance length scale of the Markov-Gaussian model (Huang and Bradford, 1992) and the limiting difference index (Paz-Ferreiro et al., 2008) were calculated. These specific indices were selected due to their common use for the quantification of soil surface roughness, as well as due to the fact that they can capture scale dependent characteristics of the soil surface. We found good agreement in the values and rainfall induced trends between all examined indices. Below we provide the major modifications we applied to the original manuscript:

Materials and Methods

A brief theory and references behind the introduced indices were added along with equations and specifics for their calculations (Pages 6-7, lines 28-11):

*"The Markov-Gaussian model is a random process that has been adopted for the quantification of soil surface roughness (Huang and Bradford, 1992; Vermang et al., 2013). In that case, the semivariogram is written as an exponential-type function with the following form:*

$$\gamma(h) = \sigma^2 \left(1 - e^{-h/L}\right), \tag{4}$$

*where $\sigma$ is the variance length scale, representing the mean roughness of a surface at the large scale, and L is the correlation length scale, which is a measure of the rate at which small scale roughness variations approach the constant value of $\sigma$. These indices are obtained by fitting the exponential-type function of Eq. (4) to the semivariogram obtained from Eq. (2).*

*Finally, the limiting difference (LD) index is another index adopted to quantify soil surface roughness. It is calculated from the first-order variogram (Linden and van Doren, 1986; Paz-Ferreiro et al., 2008), which is written in the form:*

$$\Delta Z(h) = \frac{1}{n(h)} \sum_{i=1}^{n(h)} |Z(x_i + h) - Z(x_i)|, \tag{5}$$

*Then, a linear relationship is fitted between $1/\Delta Z(h)$ and $1/h$:*

$$1/\Delta Z(h) = a + b/h, \tag{6}$$

*The limiting difference (LD) index is then calculated as $LD = 1/a$. LD has units of length, and represents the value of the first-order variance at large lag distances. It is considered as an indicator of soil surface roughness, thus adopted in the present study as an additional roughness index."*

Results

The title of the subsection 3.2 was changed to "Changes in additional roughness indices". Moreover, a paragraph was added at the end of that section, describing the findings obtained for the introduced indices (Page 9, lines 21-33):

*"Table 3 lists the Markov-Gaussian variance length scale and the limiting difference indices for the three experimental tests, and their relative change after the rainfall. These indices also show similar increase with rainfall that is of the same magnitude and trendas the RR index and crossover length, and provide a supplemental analysis about the role of rainfall intensities on the relative increase in roughness. The laser measurements from the 3 rainfall intensity*

*experimental runs were analyzed using all indices, namely, the random roughness, the crossover length, the Markov-Gaussian variance length scale, and the limiting difference indices. All indices show a consistent trend i.e., higher rainfall intensities result in higher relative increases in microroughness (Table 3).Overall, the results provided suggest that all the indices employed in this study may be used interchangeably in order to characterize rainfall induced changes in soil surface roughness, and can capture an increase in soil surface roughness, especially for low microroughness scales on the order of 2-5 mm. Our findings were compared against those reported in the literature. Huang and Bradford (1992) studied the evolution of soil surface roughness with the Markov-Gaussian variance length scale, and saw an increase of 6% in roughness for a surface of low initial roughness. Moreover, Paz-Ferreiro et al. (2008) used the LD index as an additional index to quantify soil surface roughness. They recorded a 10% increase in the LD index for a low roughness conventional tillage soil surface. Higher relative increase of roughness in our study (Table 3) compared to other studies, as seen in Fig. 5, are attributed to the significantly lower initial roughness conditions in addition to different soil types and management."*

Tables
Table 3 below was added to the manuscript, presenting our findings regarding the last two indices that were introduced in support of the RR and crossover length:

**Table 3: Summary of the rainfall induced change in the Markov-Gaussian variance length scale and limiting difference indices for the experimental tests of this study.**

| ` | Cumulative Rainfall (mm) | Pre-rainfall $\sigma$ (mm) | Post-rainfall $\sigma$ (mm) | $\sigma$ Ratio | Pre-rainfall $LD$ (mm) | Post-rainfall $LD$ (mm) | $LD$ Ratio |
|---|---|---|---|---|---|---|---|
| 30 mm/h | 150 | 1.19 | 1.63 | 1.37 | 0.79 | 0.87 | 1.10 |
| 60 mm/h | 300 | 0.42 | 1.52 | 3.62 | 0.26 | 0.87 | 3.39 |
| 75 mm/h | 375 | 0.31 | 1.43 | 4.56 | 0.15 | 0.71 | 4.84 |

**Based on the above I recommend to the editor either major revision or rejection of this manuscript.**

---

## Author Comment (AC3) · 19 Mar 2017

**RESPONSE TO REVIEW COMMENTS**

We thank the reviewer for the valuable input, which has helped improve the quality of our manuscript.   Our responses are provided below.  **Please note that the original comments are in black letters and our responses are in blue letters**.  In addition to these responses, we will provide a revised manuscript that reflect the proposed changes, as well as a copy with the tracked changes where revisions were implemented.
* * *
*General Comments:*
**In this manuscript, the authors address the effect of rainfall velocity on soil-air roughness quantified via the random roughness (RR) parameter. They showed that as rainfall velocity increased from 30 to 75 mm/hr the random roughness index increased as well, which is in contrast to those reported in the literature. Although more experimental data on support are required to have a more conclusive conclusion, the manuscript is well written and well organized and suitable for publication in the journal. However, some moderate revisions are required before publication.**

*Response:*
We thank the reviewer for the insightful comments and suggestions.  We believe that the revisions we made in response to the comments have significantly improved the quality of the manuscript.

**Minor Comments:**

**Comment 1: P3L23: Could the authors address/discuss on how changes in median diameter would affect air-soil roughness?**

*Response:*
To respond to this great question, we have utilized the median drop diameter estimated for each intensity test to calculate the terminal velocity of the rain drop (see Eq. (1d)).   Several studies have shown that soil surface redistribution under the action of rainfall is dependent on the median raindrop diameter (e.g., Warrington et al., 2009; Fu et al., 2016).  The kinetic energy (KE) corresponding to the median raindrop diameter is estimated using a collection of equations presented in Atlas et al. (1973), Begueria (2015), and Kathiravelu (2016):

$$KE_i = \frac{1}{2}m_i v_{ti}^2 \qquad (1a)$$

$$m_i = \rho_i V_i \qquad (1b)$$

$$V_i = \frac{4}{3}\pi \left(\frac{D_i}{2}\right)^3 \qquad (1c)$$

$$v_{ti} = 9.65 - 10.3 \exp(-0.6D_i) \qquad (1d)$$

where $m_i$ is the mass of raindrop i (kg), $v_{ti}$ (m/s) is the raindrop terminal velocity, $\rho_i$ is the density of raindrop i (kg/m$^3$), and $V_i$ is the volume of the raindrop i (m$^3$) which assumes a spherical shape.

In our study, the calibration of the rainfall simulators with the disdrometer allowed us to match the median raindrop sizes that are predominantly found at the study site for all 3 intensities (see Table 1.1 below).

**Table 1.1 Median drop diameters corresponding to the rainfall intensities of the experimental runs.**

| Rainfall Intensity (mm/h) | Median Drop Diameter (mm) |
|:---:|:---:|
| 30 | 2.25 |
| 60 | 2.60 |
| 75 | 2.75 |

The rate of change in relative roughness ($RR_{post}/RR_{pre}$) and the other indices against intensity (summarized in Tables 1-3 of the manuscript) essentially reflects the effects of the median drop diameter on roughness.

Eqs. (1a-d) imply that for different intensity and median drop size diameter, both the terminal velocity and mass play an important role to the RR change due to different raindrop kinetic energy. The roughness index increases with intensity; however, the relative change in roughness reduces with increasing intensity, as shown in Fig. 1.1 and summarized in Table 1.2 below (these have not been included in the paper due to space requirements). A change in rainfall intensity from 30 mm/hr to 60 mm/hr results in a 16% increase in the median drop diameter which leads to a 165% increase in the RR ratio. However, a change in intensity from 60 mm/hr to 75 mm/hr results in a 6 % increase in median drop diameter which leads to only a 29% increase in the RR ratio.

[Figure]

**Figure 1.1 Relative ratios of the roughness indices as a function of rainfall intensity.**

**Table 1.2 Increase in median drop diameter**

| Change in Rainfall Rate | Increase in Median Drop Diameter | Increase in RR Ratio | Increase in l | Increase in σ | Increase in LD |
|---|---|---|---|---|---|
| 30mm/h - 60 mm/h | 16% | 165% | 107% | 165% | 207% |
| 60 mm/h - 75 mm/h | 6% | 29% | 20% | 26% | 43% |

References

Atlas, D., Srivastava, R. C., and Sekhon, R. S.: Doppler radar characteristics of precipitation at vertical incidence, Reviews of Geophysics, 11, 1, doi:10.1029/RG011i001p00001, 1973.

Beguería, S., Angulo-Martínez, M., Gaspar, L., and Navas, A.: Detachment of soil organic carbon by rainfall splash: Experimental assessment on three agricultural soils of Spain, Geoderma 245–246, 21–30, doi:10.1016/j.geoderma.2015.01.010, 2015.

Kathiravelu, G., Lucke, T., and Nichols, P.: Rain Drop Measurement Techniques: A Review. Water 8, 29. doi:10.3390/w8010029, 2016.

**Comment 2: P4L2: The authors should clearly state that with such a low resolution some rough features with scale less than 0.5 mm have not been captured via their laser scanner. As the title indicates the authors address soil surface microroughness, while the resolution of the laser scanner is 0.5 mm. How is it possible to capture microroughness with a scanner of resolution of millimeters?**

*Response:*
We thank you for the comment. We have updated the text to include a sentence that clarifies that our analysis may not have captured microroughness features less than 0.5 mm.

We have added the following sentence to clarify the length scales that we captured (Page 4, lines 20-21):
*"Horizontal and vertical accuracies of the laser are 0.5 mm. Thus, microroughness features less than 0.5 mm may not have been captured in the analysis."*

**Comment 3: Did the authors measure infiltration rate or even saturated hydraulic conductivity of the tested soil? If so, what is the infiltration rate?**

*Response:*
In each test we placed soil moisture probes in order to continuously record volumetric water content and determine the steady-state infiltration conditions. This was roughly 2-3 hours after start depending on the rainfall intensity. We estimated the infiltration rate during all rainfall simulation runs by subtracting the measured runoff rates from the constant rainfall rates. As mentioned in the text, runoff was collected continuously at a downstream weir and rainfall rates were set to known constant value. This approach has been commonly used in plot experiments and provides a good estimate of the spatially averaged infiltration rates (e.g., Mohamoud et al., 1990; Wainwright et al., 2000). Below we provide the graph of the estimated infiltration rate with time for the 30 mm/h case:

[Figure]

Averaged saturated hydraulic conductivity ($K_{sat}$) values ranged from 3.20 – 4.56 mm/h. In our previous study, we measured $K_{sat}$ by means of semi-automated double ring infiltrometers at the field where this plot was located (see Papanicolaou et al., 2015). We found an average value of 4.0 mm/h, which is in agreement with our estimations.

The information outlined has been added to the text along with the cited references (Page 5, lines 7-12):
*"The infiltration rate was estimated during all rainfall simulation runs by subtracting the measured runoff rates from the constant rainfall rates. This approach has been commonly used in plot experiments and provides a good estimate of the spatially averaged infiltration rates (e.g., Mohamoud et al., 1990; Wainwright et al., 2000). Averaged saturated hydraulic conductivity values ranged from 3.20 – 4.56 mm/h, which are in agreement with the averaged saturated hydraulic conductivity value of 4.3 mm/h measured by Papanicolaou et al. (2015) using semi-automated double ring infiltrometers at the field where the study was performed."*

**Comment 4: P8L4-8: The authors stated that, "Analysis of soil surface roughness in the region where raindrop detachment dominates and under initial smooth surface preconditions for three rainfall intensities shows a consistent increase in the RR index and crossover length, which are confirmed as reliable descriptors of microroughness. This increase contrasts the findings of most available literature…" Please provide a few references from the literature for the last statement.**

*Response:*
Per the suggestion of the other referees, we have removed the last statement because our findings do not contradict, but rather complement the existing literature by covering a range of initial microroughness that has not been captured before. That part of Discussions and Conclusion section now reads as follows, with references added to support it (Page 10, lines 5-10):

*"The results obtained are consistent with findings of other studies that have examined length scales up to 5 mm (Burwell et al., 1963; Allmaras et al., 1966; Burwell et al., 1969; Cogo, 1981; Currence and Lovely, 1970; Steichen, 1984; Unger, 1984; Zobeck and Onstad, 1987). These length scales (i.e., ~2-5 mm) have been found to be common in agricultural landscapes due to prolonged exposure to rainfall impact, runoff and freeze-thaw cycles. Within these landscapes, the reported increase is expected to occur during the early part of the storm where rainsplash action may be more important than runoff."*

**Comment 5: Did the authors measure soil aggregate- or particle-size distribution? What is the range of particle sizes in mm?**

*Response:*
Yes, the aggregate size distributions of the soil studied were measured. We found 19% of the soil size fraction less than 250 μm, 48% between 250 μm and 2 mm, and 33% greater than 2 mm.

We have added the aggregate size distribution of our soil along with other info for a clearer presentation (Pages 3-4, lines 29-1):
*"The soil series at the plot where the experiments were conducted is Tama (fine-silty, mixed, superactive, mesic Cumulic Endoaquoll) (http://criticalzone.org/iml/infrastructure/field-areas-iml/). It consists of 5% sand, 26% clay, 68% silt, and an organic matter content of 4.4%. The aggregate size distribution of the soil consists of 19% of the soil size fraction less than 250 μm, 48% between 250 μm and 2 mm, and 33% greater than 2 mm. These soils contain both smectite and illite, with high cation exchange capacity between 15 and 30 Meq/100 g."*

---

## Author Response (AR2)

**RESPONSE TO REVIEW COMMENTS**

Dear Professor Daniel Schertzer,

We have reviewed the comments you provided very carefully. We want to thank the reviewer for the in-depth comments, suggestions and corrections, which overall have greatly improved the overall quality of the manuscript. Our responses are provided below. **Please note that the original comments are in black letters and our responses are in blue letters**. In addition to these responses, we will provide a revised manuscript which reflects the proposed changes, as well as a copy with the tracked changes where revisions were implemented.
* * *
**General Comments:**

**The revised version of this manuscript has been very much modified regarding the first version. Several points that were raised by reviewers and editor have been addressed. Thus, organization has been somewhat improved, complementary information has been provided in the material and methods section, new indices have been taken into account and a number of sentences have been rephrased or corrected in all the sections of the manuscript.**

**However, in my opinion, this manuscript is not acceptable in its present form and therefore I still recommend major revision or rejection. This is mainly because results presented here are only significant for a very particular situation, but in spite of this generalization without experimental support have been made. Subsequently, presentation of results (including results cropped from the literature) is neither clear nor concise; also results are not put in context of the previous existing experimental evidence and text is far from precise.**

**First, I recognize that the results provided are interesting. However, the intended generalization from experimental results of a particular or "unique" case is one of my main concerns. Therefore, I completely agree with a previous comment of a different reviewer, which stated: "(I have) the feeling that authors overestimated the relevance of their results and reached conclusions that are not sufficiently proved by their data". In my opinion this statement still remains absolutely valid for the current version of the manuscript. For example, several statements in the Abstract section are too vague, imprecise or even misleading. Also other statements in the Discussion and Conclusions section are too strong or even not supported by results and evidence**

*Response:*

We thank Reviewer #2 for their comments and noting the improvements made. We acknowledge the concerns and have made every effort to remove any overestimations, generalizations, and conclusions that are not based on the results of the specific conditions examined in the study. We have made it clear in the paper, based on the results, where our opinions point toward the need for further investigations. Please see answers to the Specific comments below that addresses these (Abstract: Comment 4, Introduction: Comment 2, Results: Comment 4, Discussions and Conclusion).

**Second, preparation of soil surface to obtain a smooth surface in this work could result in an important bias. To the best of my understanding "tamping using a plywood board" for smoothing is an uncommon method. Using such procedure compaction of soil aggregates and structural units should be expected. Compaction intensity should increase with increasing soil water content and strength of tamping. Subsequently, the evolution of small aggregates, resulting mainly from tamping, under simulated rainfall could be far away from that of aggregate beds prepared by methods commonly used in field and laboratory experiments.**

*Response:*
We followed the tamping approach in order to have consistent initial conditions for the replicates we performed out in the field. This was so that if any bias was introduced it would be consistent, if not minimal, and would not affect the trends in the experimental results. We do acknowledge that there could have been some compaction of soil aggregates and some effects from the compaction intensity. This could be problematic if soil aggregation was the focus of this research. However, this is not the case here. We strongly believe that the increasing trend in roughness observed in this study did not arise due to tamping and was due to the initial smooth conditions. Furthermore, our efforts to ensure consistency is confirmed in the values of all roughness indices for the pre- and post-rainfall roughness, which are consistent both in trends and relative magnitudes. This is further addressed in our answers to the Specific comments below (Abstract: Comment 4, Introduction: Materials and Methods: Comment 1, Results: Comment 4). We also acknowledge the potential impact of compaction of soil aggregates in the Discussions section of the manuscript.

**Specific Comments:**

**Abstract**

**Comment 1: Page 1, Lines 13 to 17. On the one hand it is stated that roughness of about 2-5 mm is common on agricultural landscapes, but on the other hand the focus of this**

**manuscript is on roughness of 2 mm or less. I don't understand the discrepancy between the problem to be dressed and the focus of the manuscript.**

*Response:*
Thank you for pointing this out. We understand the confusion stemming from how the sentences were framed. Our motivation was to study length scales less than 5 mm, where increases in roughness have been reported (e.g., Vázquez et al., 2008; Zheng et al., 2014). Since most of the studies for these length scales had roughness ranging between 2-5mm, we chose 2 mm as our reference condition for our experiments (since it was the apparent lower limit). However, ensuring an exact roughness value of 2 mm in the field is difficult, hence the slightly less initial roughness values. The initial part of the abstract has therefore been modified as follows (Page 1, Lines 13-19):

*"This study examines the rainfall induced change in soil microroughness of a bare smooth soil surface in an agricultural field. The majority of soil microroughness studies have focused on surface roughness on the order of ~5-50 mm and have reported a decay of soil surface roughness with rainfall. However, there is quantitative evidence from few studies suggesting that surfaces with microroughness less than 5 mm may undergo an increase in roughness when subject to rainfall action. The focus herein is on initial microroughness length scales on the order of 2 mm, a low roughness condition observed seasonally in some landscapes under bare conditions, and chosen to systematically examine the increasing roughness phenomenon."*

**Comment 2: Page 1, Line l5. What means "long undisturbed exposure to rainfall impact"?.**

*Response:*
This phrase was removed to avoid confusion.

**Comment 3: Page 1, Line 18. "generic extreme conditions" is too vague. Also, it is not clear, here and all over the paper if your focus is in extreme conditions or in conditions that are common in agricultural landscapes.**

*Response:*
This phrase was removed. Now it is explicitly stated that the focus of this paper is on initial microroughness of the order of 2 mm.

**Comment 4: Page 1, Lines 25 to 27. The statement beginning "this contradicts..." and ending "… roughness conditions" is awkward, as before noted by other reviewer. What is the meaning of "monotonic" in this context?. The "commonly adopted literature" (for examples Zobeck and Onstad, 1987, who analysed 483 data sets and since them several other works) is the result of hundreds, perhaps thousands of experimental evidence in**

**different soils with different roughness magnitude, texture, aggregate stability, soil water content etc. In contrast in the present work results are limited to three soil surfaces from a particular soil, and the results obtained may be biased by the soil preparation procedure.**

*Response:*
Thank you. We acknowledge the awkwardness of the statement and have modified the sentence. Our intent was not to suggest a limitation of the other studies but merely to point out that there are some conditions, specifically smooth surfaces, which have not yet been examined in detail, hence the limited data. Our study is a first step towards acknowledging this gap and filling it. Whilst we acknowledge that the conditions we examine are limited, we do not agree that the trends we observed are biased or insignificant based on the soil preparation procedure as other studies with a different soil preparation procedure have also reported an increase in microroughness e.g., Vázquez et al. (2008) and Zheng et al. (2014) (see earlier response to your general comment).

We have modified our statements in response to the comment to convey this message by indicating that the outcome of the interaction between rainfall and a soil surface may be different for smoother surfaces comparatively to rough surfaces, and highlights the need for a better understanding of the interaction (Page 1, Lines 22-26):

*"Findings show a consistent increase in roughness under the action of rainfall, with an overall agreement between all indices in terms of trend and magnitude. Although this study is limited to a narrow range of rainfall and soil conditions, the results suggest that the outcome of the interaction between rainfall and a soil surface can be different for smooth and rough surfaces, and thus warrant the need for a better understanding of this interaction."*

**Etc., etc., etc. Again, my comments are not exhaustive. Altogether, I feel this abstract as awkward.**

**Introduction**

**Comment 1: Page 2, Lines 10 to 21. Indeed, random roughness assessment requires correction for both, slope and tillage marks. This is clearly stated in lines 18 and 19, but not in lines 10 and 11. Text in Lines 10 to 21. Also, please consider if Equation 1 should be in the Methods section.**

*Response:*
Now it is clearly stated that evaluation of the RR index was done after correction for both slope and tillage marks in the Introduction (Page 2, Line 7).

Equation (1) was moved to the Methods section (Page 5, Lines 14-19).

**Comment 2: Page 2, Lines 28 to 29. Please, note that Kamphorst et al. (2000) analysed 48 seedbeds with a roughness of the order of magnitude that you mention. Again 48 is much higher than 3, and several different soils have been studied by Kamphorst et al. (2000). Therefore, even if your results may be of interest, the should be put in the context of the previous work about the studied topic, and care should be taken to avoid overestimation of their relevance.**

*Response:*
We have modified the text, explicitly stating that other studies have examined soil surface roughness <5 mm. We have also removed the extrapolation to limitations of hillslope scale erosion models in regards to representation of storage, runoff and sediment routing. We however note that there is no contradiction between the Kamphorst et al. (2000) results and our findings.

We state explicitly in the manuscript that we focus on the quantification of rainfall induced change in soil microroughness for scales 2 mm or less to examine the potential increase in roughness as the evidence suggests. This focus is due to limited data available for examining this trend and the lack of acknowledgement of the trend in previous studies (Page 2, Lines 15-29):

*"There are few studies that have examined surfaces with initial microroughness less than 5 mm, a low roughness condition observed seasonally in some landscapes under bare conditions (e.g., Kamphorst et al., 2000; Vázquez et al., 2008; Zheng et al., 2014). Hereafter, for shortness, tests with initial RR less than 5 mm will be referred to as "smooth", whereas tests with initial RR greater than 5 mm will be referred to as "rough". There are some quantitative indications that under bare smooth surface conditions, soil surface roughness may actually increase under the action of rainfall… However, none of the… studies explicitly stated or acknowledged the increasing trend of roughness and its potential linkage to smooth bare soil surface conditions. The main goal of this study is to examine changes in RR under rainfall impact for initial microroughness less than 2 mm, since this appears to be the lower limit of roughness scales examined in the literature."*

Special care is taken throughout the text so as not to overestimate the relevance of our results.

**Comment 3: Page 2, Lines 29 to 34 and page 3, Lines 1 to 2. Sentences beginning with "Surfaces with microroughness…" are too vague. In addition, please note that in conventional tillage between postharvest and plant growth there is seedbed preparation. In**

these conditions seedbeds constituted by small aggregates are the best example of real "microroughnees" or smooth soil surface.

*Response:*
We agree that the statement may come across as vague. These sentences were removed to improve the clarity and precision of the manuscript.

**Comment 4: Page 3, lines 12 to 16. Again, I don't understand that if the aim of the work is to address the microrelief in soil surfaces with aggregates of about 2-5 mm you are focusing on "the order of 2 mm o less".**

*Response:*
The goal of this work has now been corrected to address rainfall effects on roughness at scales on the order of 2 mm (Page 2, Lines 28-29). As explained above, the observation from literature review that surfaces with microrelief < 5 mm can undergo an increase in roughness under rainfall and the lack of data for smooth conditions is the main motivation of the study. We acknowledge the narrow range of our studied conditions, yet we believe our study provides the first leap for further exploring the behavior of smooth surfaces, and motivates further research on how the interaction between a soil surface and rainfall would be affected by the initial roughness conditions, and which are the underlying physics.

**Comment 5: Page 3, lines 19 and 20. Although "disturbed" nay be an antonym of "smooth", in the context of this work correct is "rough". Please, see also Table 1.**

*Response:*
The work "disturbed" has been replaced by "rough" throughout the manuscript.

**Comment 6: Page 3, line 22. "Different intensities" of which?**

*Response:*
That part of the text was corrected to "different rainfall intensities".

**Etc., etc., etc. Again, my comments are not exhaustive. Again, tightening up is required.**

*Response:*
Done – the manuscript has been tightened up.

**Material and Methods**

**Comment 1: Page 4, line 4. What about aggregate size distribution after tamping?. What about soil compaction before and after tamping? What about aggregate stability before and after tamping?. Aggregate stability is a very important issue in this particular soil, prepared with a particular procedure before laser scanning. This is because of the strong relationship between processes responsible for roughness and aggregate stability.**

*Response:*

It is recognized that the soil preparation method in our study could have introduced some bias to the soil properties such as aggregate size distribution, compaction, and aggregate stability. Nonetheless, for the purpose this study was designed for, this preparation method ensured consistency in the initial and final roughness states, as confirmed by replications of our experimental runs. Therefore, the bias introduced to our results would be consistent, if not minimal. In addition, other studies with a different soil preparation procedure where aggregates were not potentially disturbed and the conditions mimicked natural soil surfaces have also reported an increase in microroughness (e.g., Vázquez et al., 2008; Zheng et al., 2014). This suggests that the increasing trend in surface roughness is not an "artifact" of our soil preparation procedure.

The text has been modified as follows to acknowledge the potential existence of a consistent bias introduced by our preparation method (Page 3, Lines 11-14):

*"The soil surface was prepared before each experiment by tamping using a plywood board to create a smoothened surface. This was done to ensure a consistency in surface roughness between the experiments, as well as to ensure that any potential bias introduced in the plot preparation would be also be consistent, if not minimal. This was confirmed by the observed roughness of the experiment replicates..."*

**Comment 2: Page 5, Line 11. Average hydraulic conductivity was rather low compared to simulated rain intensity. This is very important, since in these conditions in a flat surface should be expected. What about water ponding in your experiment, given that your plots had a slope?. Ponding is known to interfere with roughness decay.**

*Response:*

Ponding on the soil surface of the experimental plots within the region of interest was minimal throughout the experiments. This can be seen in the image below, which was taken at the later part of a simulated event. The minimal ponding is attributed to the smooth bare surface

conditions and the high plot gradient of 9%, leading to low depressional storage. The above statement was added to enhance the precision of the text (Page 4, Lines 23-26):

*"Although the average saturated hydraulic conductivity values were low with respect to the applied rainfall rates, minimal ponding was observed on the experimental plot, owing to the smooth bare conditions and the high plot gradient of 9%, which led to low depressional storage"*

[Figure]

**Comment 3: Page 5, Lines 15 to 18. The difference in initial roughness between experiment 1 and experiment 2 and 3, demonstrate roughness decay in your soil under natural rainfall. This is against your main hypothesis, which postulate roughness increase with increasing rainfall.**

*Response:*
As it is stated in the text, the plots were located in an actual agricultural landscape and the plot for Experiment 1 had recently been disturbed by tillage, while Experiments 2 and 3 were performed later in the season. In between these periods, the effect of runoff from upslope areas is considered to have contributed to the decay in roughness over the prolonged period for which the plots for Experiments 2 and 3 were exposed. Therefore, roughness decay under natural rainfall was primarily because of the effect of runoff, and not the actual rainfall detachment process which is investigated in our study. The location of the plot in relation to the hillslope is shown on the map below to highlight this. This map has also been included in the manuscript as Figure 1. To avoid confusion regarding the decay, the statement below was added to the text (Pages 4, Lines 27-32).

*"The initial microroughness length scale in Experiment 1 (1.17 mm) was greater than that of Experiment 2 (0.42 mm) and Experiment 3 (0.32 mm) – see Table 1. This is attributed to the different timing the experiment runs were performed with respect to tillage. Experiment 1 was performed in early August, soon after harvest, so the soil surface had recently been disturbed. However, for Experiments 2 and 3 which were performed in late September, the soil presented less surface disturbance due to the cumulative action runoff from upslope areas on the plots arising from natural rainfall within that period (Papanicolaou et al., 2015b)."*

[Figure]

**Location of experimental plot in relation to hillslope. The figure shows that the plot receive runoff from upslope areas of the hillslope.**

**Comment 4: Page 5, Lines 25. You stated that "no rill formation ever took place". But, what above "micro rills"?. I'm not sure above the absence of micro rills in your experimental conditions. Indeed, "traditional" works about soil surface roughness decay absolutely exclude the presence of soil erosion.**

*Response:*

Visual evidence from the experiments confirms that rainsplash was the dominant process within the top part of the plot where our regions of interest were located (see figure provided in comment 2) and that the presence of micro rills was minimal. Given the very small flow rates at this location, even if microrills were present, the low stream power associated with their function would lead to little contribution to the alteration of soil surface roughness compared to the action of raindrop detachment (Kinnel, 2005). Furthermore, Kinnel (2005) noted that the impact of microrills is more prevalent for slopes gradients beyond 10%. We thus believe that our results were not affected by micro rills.

**Comment 5: Page 3, Lines 3 to 24. I wonder if semivariograms, Hurst exponent, etc, have been calculated after removal of oriented roughness, i.e. tillage marks and also slope. Etc., etc., etc. Again, my comments are not exhaustive. Again, tightening up is required.**

*Response:*

Yes, correction for both slope and tillage marks was performed before the estimation of all the microrelief indices. This has also been added to the text where the various indices are described.

**Results**

**I'm very concerned by the procedure used to compare results obtained in this work have with those of previous works. Results shown in Table 1 and in Figure 5 are misleading.**

**Comment 1: First, as before stated, hundreds of soil surfaces have been analysed for roughness decay in very different experimental conditions and this work only provide results for three soil surfaces measured under "unique" experimental conditions, which**

**perhaps lead to biased results. Moreover, you should include also the data you obtained showing roughness decay under natural rain conditions; this is given by the difference between initial roughness in Experiment 1, performed in earlier August (1.17 mm) and in Experiment 2 and 3, performed in late September (0.42 mm and 0.32 mm). This clearly show roughness decay in your soil with increasing natural rainfall isn't it?.**

*Response:*

As discussed in Comment 3 of the Material and Methods section above, the decay in the initial roughness for Experiments 2 & 3 was influenced by runoff due to the position of the experimental plot on the landscape. As noted, Experiments 2 & 3 were performed at a later time of the year than Experiment 1. However, this difference in initial roughness is not an issue since the results are presented in a dimensionless form, i.e. the random roughness ratio ($RR_t/RR_0$; where $RR_t$ is the random roughness after rainfall application and $RR_0$ is the initial random roughness).

**Comment 2: Second, I don´t understand the criteria for selecting data from previous authors. In your answers to reviver's you stated that you only provide data for one simulation in order to focus on the analysis of raindrop impact. However, most experiments quoted in Table 1 and Figure 5 includes several successive events of natural or simulated rainfall. In addition, the selected data sets correspond to very heterogeneous initial conditions, texture, organic matter content, rainfall application, etc.**

*Response:*

We acknowledge that because of the heterogeneous conditions of the experiments cited, Figure 5, and Tables 1 and 2 may present some confusion and can appear as misleading. Therefore, Figure 5 has been removed from the manuscript, while the Tables 1-3 have been modified. Table 1 now only includes results of our study along with results from the two studies focused on smooth initial surface conditions and reported an increase in roughness. With these changes, although the experimental conditions may still differ in Table 1, they do refer to smooth surfaces and similar soils, which support the increasing trend due to rainfall. Table 2 was merged with Table 3, presenting only results of our study with respect to alternative microroughness indices. The texts accompanying the Tables have also been modified accordingly.

**Comment 3: Third, I don't understand the way in which calculations of RR ratio have been performed. For example, Vazquez et al (2007, 2008) quoted several measurements of RR for a given surface and you selected only one of them, apparently the last one. Why?.**

*Response:*

The main goal of our study is to assess the steady-state conditions of roughness in order to get comparable results for different rainfall intensities. Therefore, we calculated the RR index between initial conditions and most final conditions in each experiment. For the experiments by other investigators (Vazquez et al., 2008) that examined a succession of events, we selected the roughness after the final succession since this was deemed as more closer to steady state conditions and, thus, more comparable to our study. With regards to the Vazquez et al (2008) study, selection of the intermediate storms relative the initial roughness would still have shown an increase in roughness for two of the three experiments that were performed. Only one of the experiments was an exception. We do agree with the reviewer however that a better understanding of the role of rainfall succession is needed in subsequent experiments. To put the studies we present into context, we elaborate further on the selection of the final roughness and the measurements and trends in the other studies (Page 7, 18-25):

*"Table 1 summarizes the results of this study along with results from other studies focused on smooth surfaces, documenting the RR index values before and after the rainfall events, the cumulative rainfall, as well as the associated RR ratio. The present study, along with Vázquez et al. (2008) and Zheng et al. (2014) generally report an increase in RR with rainfall under the conditions examined. The Vázquez et al. (2008) study, however, differs from the present study and Zheng et al. (2014) in that it examined roughness evolution under successive rainfall events per run. Only the RR data collected on completion of the last rainfall succession in each run are presented in Table 1. The final RR values after the last rainfall succession were selected for being the more closely comparable to the steady-state conditions examined herein."*

**Comment 4: Summarizing, I'm sorry, but I have no evidence showing an effect of initial RR condition on the RR ratio. The interesting apparent exception found in your work may be the result of experimental biases and may be rather the result of an "artefact".**

*Response:*
This comment has been addressed in our responses above. We think the issue of the bias has been addressed since there are several supporting evidence that this is not an artifact.

**Comment 5: On the other hand, I have positive comment regarding subsection 3.2 (pages 8 and 9). Points raised here are important to further assess RR.**

*Response:*
Thank you. We have overall tried to highlight more these points and the interesting findings of our work, avoiding generalizations and only stating what our results would imply and the needs for further investigation.

**Discussion and Conclusions**

**Comment: I can't review this section before the above mentioned points are addressed. However, please let me comment that the expected hydrological impact of a soil surface roughness in the order of 0.5 to 1 or 2 mm would be rather scarce. This is because of the expected digressional storage would be less than 1 mm per quadrat meter (Kamphost et al., 2000)**

*Response:*

The impact of microroughness generation after a rainfall event can have a significant hydrological effect based on other factors besides digressional storage. As an example, we demonstrate this impact with a well-established pedotransfer function for the effects of soil crusting, roughness, and rainfall kinetic energy on the bare hydraulic conductivity, $K_{br}$, (Risse et al., 1995) that is employed in some overland flow models (e.g. WEPP; Flanagan *et al*., 1995). The relationship is expressed as follows (Risse et al., 1995):

$$K_{br} = K_b \left[ CF + (1 - CF)e^{-C.E_a(1 - RR_t/RR_{t-max})} \right] \qquad (1)$$

where $K_b$ is the baseline hydraulic conductivity, *CF* is the crust factor, *C* is soil stability factor, *Ea* is the cumulative rainfall kinetic energy since the last tillage, $RR_t$ is random roughness height, and $RR_{t-max}$ is the maximum random roughness height. Using the following typical values for our study site based on literature (Flanagan *et al*., 1995; Chang, 2010): Ea = 10,000 J/m²; C = 0.0002 m²/J; $RR_{t-max}$ = 40 mm, we present the plot below for an initial $RR_t$ value of 2 mm and minimal *CF* factor. The plot shows the percentage change in the bare hydraulic conductivity with increasing random roughness (RR) ratio. For the highest ratio observed in this study (4.5), we note that the percentage change in hydraulic conductivity can be as high as 42%. A paragraph has now been incorporated into the manuscript to present this potential impact.

[Figure]

Another example of where this roughness scale is important is on the mobilization of finer soil fractions and the estimation of the enrichment ratio (ER) for determining nutrient fluxes. The relationship between the change in roughness and the dislodgement/transport of finer sized fractions is poorly understood and for which a more detailed understanding is needed since the ER has been found to be particularly sensitive under the action of rainsplash (Papanicolaou et al., 2015). This factor has not been accounted for before and there is a need to build on our findings herein.

Per the comments brought up in the other sections related to generalization and overestimation of the results, the following sections have accordingly been removed from the Discussion section:

Page 11, Lines 12-13:
*"It is further demonstrated that for low microroughness scales the relative increase in roughness increases with rainfall intensity."*

Page 11, Lines 15-21:
*"Increase in microroughness further infers increase in depression storage at the soil surface prior to runoff generation (Kamphorst et al., 2000), which can affect ponding and flow pathway patterns especially at the onset of a storm event (Onstad, 1984). The results obtained are consistent with findings of other studies that have examined length scales up to 5 mm. These length scales (i.e., ~2-5 mm) have been found to be common in agricultural landscapes that are subject to prolonged exposure to rainfall impact, runoff, and freeze thaw cycles (Burwell et al., 1963; Allmaras et al., 1966; Burwell et al., 1969; Cogo, 1981; Currence and Lovely, 1970; Steichen, 1984; Unger, 1984; Zobeck and Onstad, 1987)."*

Page 12, Lines 14-15:
*"Nonetheless, the current findings may help explain some modeling discrepancies in terms of depression storage and runoff predictions."*

*Page 12, Lines 16-26:*
*"On an annual basis, Abaci and Papanicolaou (2009) and Abban et al. (2016) highlight the importance of the seasonal variation of land cover on sediment output in agricultural Intensively Managed Landscapes (IMLs), indicating that during certain periods, the combination of high magnitude events and bare soil will severely increase erosion. This point is of relevance here given the soil surface in agricultural IMLs is bare 30-75% of the time during the calendar year. Models simulating these periods at the microscale are likely to be sensitive to the treatment (and definition) of the soil surface microroughness, and thus, require an adequate determination of the soil surface roughness length scales for accurately modeling the hydrologic response of hillslopes. To the extent that microscale processes are considered significant, we argue that such*

*models should adequately capture the increasing and decreasing trends in soil microroughness during all stages of a storm event in order to accurately predict local response to rainfall. The extent to which the increase in RR recorded herein can affect erosion processes is not yet known. However, it has been noted that different values of RR can affect flow pathways and runoff, which consequently can affect erosion processes (Gómez and Nearing, 2005)."*

Page 12, Lines 27-32:
*"The majority of existing models assume that RR always decays over time with rainfall. Few models consider the reverse condition where the soil surface is initially smooth as defined in the current paper and RR increases under the action of raindrop. By providing the ratios of increase in roughness indices with rainfall intensity, the parameterization of the evolution of surface roughness with rainfall could be improved for current models. Future research will provide a better understanding of the extent to which the initial increase in roughness in the early part of the storm could have an impact on flow pathways, runoff, and processes at subsequent parts of the storm."*

15   . This, however, is not an issue since all the results are presented herein in a dimensionless form (see Section 2.2 below on the index ratios). All cases, nonetheless, exhibited initial microroughness length less than 2 mm corresponding to smooth surface bed conditions as confirmed with the laser scanner. Dry soil bulk density was 1.25 g/cm$^3$ for Experiment 1, and about 6% higher for Experiments 2 and 3 due to self-weigh consolidation of soil.

20   Figure 5a provides an example of the experimental plot at pre-rainfall and post-rainfall conditions. Since the focus of this research is only on plot regions where raindrop detachment is dominant over runoff, we are using the scanned profiles that correspond only to these upslope locations, which are shown in Fig. 5b. Rill formation was not observed in these regions throughout the experiments. Visual observations confirmed that raindrop detachment was dominant and the main driver of the change in soil surface

[revised manuscript text omitted]
  that existed in the initial microrelief amongst the three runs due to the different timing of the experiments (see Section 2.1), and compare rainfall-induced changes in relative terms, the results from the rainfall experiments are presented in the form of ratios of the roughness indices. More precisely, the RR ratio, defined as the ratio of the RR index post-rainfall over the RR index prior to the rainfall ($RR_{post}/RR_{pre}$), is calculated for each experiment. Semivariograms are plotted under pre- and post-rainfall conditions at the ROI to assess the spatial correlation of surface elevations. Along the same lines, ratios between pre- and post-rainfall conditions are calculated for the crossover length, the variance length scale of the Markov-Gaussian model, and the limiting difference to assess changes in microroughness along with the RR ratio.

**3. Results**

**3.1 Changes in the RR index**

Based on visual inspection of the DEMs in Fig. 5b, it is evident that microroughness in the splash-dominated region increases with rainfall. ~~Figure 5 shows the RR ratio, i.e., $RR_{post}$/$RR_{pre}$, with respect to the initial value of RR for the present study along with other studies that quantify rainfall induced microroughness changes. The dashed line at the RR ratio value of unity reflects no change in roughness, thus all points above that line show an increasing trend with rainfall, while all points below show a decreasing trend with rainfall. All the studies capture a wide range of initial RR values – up to 21 mm – and it is clear that our study captures the behavior of RR for an initial range that was not covered before. Figure 5 suggests that roughness may increase with raindrop impact for a range of low initial RR values (< 5 mm), while it consistently decays for high initial RR values (> 5 mm). It is acknowledged that the values of the roughness indices among different studies may reflect different conditions such as rainfall forcing and soil type. For example, Vázquez et al. (2007) used clay textured soil, Vázquez et al. (2008) used silt loam textured soil, while our study along with all the other studies cited conducted rainfall experiments for silty clay loam textured soil. Rainfall intensities and cumulative rainfall amounts varied significantly among studies.~~

Table 1 summarizes the results of this study along with results from  other studies focused on smooth surfaces, documenting the RR index values before and after the rainfall events, the cumulative rainfall, as well as the associated RR ratio. The present study, along with Vázquez et al. (2008) and Zheng et al. (2014) generally report an increase in RR with rainfall under the conditions examined. The Vázquez et al. (2008) study, however, differs from the present study and Zheng et al. (2014) in that it examined roughness evolution under successive rainfall events per run. ~~Exception seems to hold for one soil surface of the study of Vázquez et al. (2008), as well as the smooth surfaces of Vermang et al. (2013) which shows decaying roughness due to rainfall because of different soil type and rainfall conditions. Note that Second, the present study indicates that the RR ratio becomes higher with higher rainfall intensity when the surface is classified as smooth, whereas the opposite tends to hold for soil surfaces classified as disturbed (Fig. 5, Table 1). Vázquez et al. (2008) performed successive rainfall simulations and recorded the values of roughness for each succession:in the present study the last successionthe calculation of the final state of RR,–butwe didmay be attributedtheorsthat they either appliedorin their study were higher~~different.

Other studies not included in Table 1 have also shown increasing trends of roughness with rainfall, as quantified with the use of different indices. For instance, Huang and Bradford (1992) calculated the semivariograms for different surfaces and used fractal and Markov-Gaussian parameters to quantify the roughness. Markov-Gaussian analysis showed a relative increase in the roughness parameter for a surface of low initial roughness. Finally, Rosa et al. (2012) introduced the Roughness Index, which is estimated from the semivariogram sill (i.e., the upper value where the semi-variance levels out), in order to quantify roughness, and observed  an increase with rainfall under low initial roughness conditions. That increase was attributed to the fragmentation of aggregates and clods to smaller aggregates but was not linked to smooth bare soil surface conditions. Overall, the experimental evidence suggests that the interaction between rainfall and smooth soil surfaces can lead to an increase in microroughness.

The results outlined above for the use of the RR index as a descriptor of change in microroughness have been based on the assumption that there is no statistically significant spatial correlation in elevation readings between neighboring locations at the ROI. This condition was indeed not violated due to the choice in ROI.  The following subsection outlines and discusses the results of the semivariogram analysis and additional indices used to confirm the validity of  the assumption and  comparison with the RR index method.

**3.2 Changes in alternative roughness indices**

[revised manuscript text omitted]

**4. Discussion and Conclusions**

Many studies have examined the response of rough surfaces to rainfall, and have reported a decay of roughness. Few studies have been developed to assessed microscale variation of smooth surfaces in response to rainfall under controlled conditions to purposely examine increase in RR with rainfall intensity. Unique The experiments are presented herein that were designed to help us decipher the role of rainsplash on increasing RR for smooth surfaces with initial microroughness below on the order of 2 mm by isolating the role of other processes factors such as runoff, variable water content, bare soil surface, and soil texture, among others etc. Our results show a consistent increase in roughness under the action of rainfall, with an overall

agreement between all the roughness indices examined herein in terms of trend and magnitude. Our findings are consistent with findings of other studies that have examined length scales less than 5 mm and  may suggest the possible existence of a characteristic roughness  threshold  below which RR is expected to increase  due to the action of rainfall. The value of this threshold  may depend on the specific soil and rainfall conditions. A caveat of our study is that due to the limited range of conditions examined herein more experiments are needed to further solidify the conditions under which RR is expected to increase under rainfall action. An outcome of this study is the ~~fact that the mere action of rainfall cannot completely smoothen out a bed soil surface, thereby localized microroughness residuals will always remain at the locations where the action of runoff is low or absent. Increase in microroughness further infers increase in depression storage at the soil surface prior to runoff generation (Kamphorst et al., 2000), which can affect ponding and flow pathway patterns especially at the onset of a storm event (Onstad, 1984). The results obtained are consistent with findings of other studies that have examined length scales up to 5 mm. These length scales (i.e., ~2-5 mm) have been found to be common in agricultural landscapes that are subject to prolonged exposure to rainfall impact, runoff, and freeze-thaw cycles (Burwell et al., 1963; Allmaras et al., 1966; Burwell et al., 1969; Cogo, 1981; Currence and Lovely, 1970; Steichen, 1984; Unger, 1984; Zobeck and Onstad, 1987). Wwthesethe reportedis expectedmay beThe exact mechanisms leading to increase in roughness are unknown. Changes in roughness during a storm event can be attributed to compression and drag forces from the raindrop impact on the soil, angular displacement due to rainsplash, aggregate fragmentation, and differential swelling (Al-Durrah and Bradford, 1982; Warrington et al., 2009; Rosa et al., 2012; Fu et al., 2016).~~

Regions exhibiting different median raindrop diameters

This study suggests that the effects of the interaction between rainfall and a soil surface can be different for smooth and rough surfaces, and highlights the need for a better understanding of the interaction due to its potential impact on hydrologic response. This potential impact is demonstrated

with the following established pedotransfer function for the effects of soil crusting, roughness, and rainfall kinetic energy on the bare hydraulic conductivity, $K_{br}$, (Risse et al., 1995):

$$K_{br} = K_b\left[CF + (1 - CF)e^{-C.E_a(1-RR_t/RR_{t-max})}\right] \tag{7}$$

where $K_b$ is the baseline hydraulic conductivity, $CF$ is the crust factor, $C$ is soil stability factor, $E_a$ is the cumulative rainfall kinetic energy since the last tillage, $RR_t$ is random roughness height, and $RR_{t-max}$ is the maximum random roughness height. Using the following typical values for the study site based on literature (Flanagan *et al.*, 1995; Chang, 2010): $E_a = 10,000$ J/m², C = 0.0002 m²/J, $RR_{t-max} = 40$ mm, the percentage change in bare hydraulic conductivity for increasing roughness can be estimated for an initial $RR_t$ value of 2 mm and minimal $CF$ factor. Performing the analysis for the range of random roughness ratios observed in this study (~1.3 − 4.5), the percentage increase in hydraulic conductivity is found to range between 5% −42%, which will have a significant impact on rainfall-runoff partitioning.

It is recognized that the soil preparation method in our study could have introduced some bias to the soil properties such as aggregate size distribution, compaction, and aggregate stability. Nonetheless, for the purpose this study was designed for, this preparation method ensured consistency in the initial and final roughness states, as confirmed by replications of our experimental runs. It is also recognized that drier, silty type soils may not exhibit the increase in RR shown here. Further, the role of sealing may be important on roughness development under bare soil conditions and needs further examination. Soil water retention characteristics of the soils under sealing and its implication to RR must be considered (Saxton and Rawls, 2006). Finally, the role of successive storm events on changing roughness for smooth surfaces is not covered in this study and needs to be examined.

The exact mechanisms leading to increase in roughness remain unknown and are not the focus of this study. However, changes in roughness during a storm event have been attributed to compression and drag forces from the raindrop impact on the soil, angular displacement due to rainsplash, aggregate fragmentation, and differential swelling (Al-Durrah and Bradford, 1982; Warrington et al., 2009; Rosa et al., 2012; Fu et al., 2016). Regions exhibiting different median raindrop diameters may experience different soil surface roughness evolution due to different aggregate fragmentation and rain splash effects (Warrington et al., 2009; Rosa et al., 2012; Fu et al., 2016). Future research should explore these mechanisms.

and potential implementation in s  always ~~be the case. Finally, this study and other studies demonstrate that the evolution of soil surface roughness in response to rainfall is dependent on initial roughness conditions and can contribute to hydrology, i.e., another factor shaping the soil surface (e.g., through runoff). Different behavior of surface roughness evolution, i.e., increase or decrease, depending on initial roughness conditions indicates a dynamic and nonlinear feedback between hydrologic response and surface roughness which may affect depression storage, ponding and flow pathways (Kamphorst et al., 2000; Gómez and Nearing, 2005). However, the extent to which soil surface roughness increase would affect depression storage, ponding, and flow pathways is unknown, and although one would expect it to be minimal, more research is needed to quantify it under different conditions.~~

~~On an annual basis, Abaci and Papanicolaou (2009) and Abban et al. (2016) highlight the importance of the seasonal variation of land cover on sediment output in agricultural Intensively Managed Landscapes (IMLs), indicating that during certain periods, the combination of high magnitude events and bare soil will severely increase erosion. This point is of relevance here given the soil surface in agricultural IMLs is bare 30-75% of the time during the calendar year. Models simulating these periods at the microscale are likely to be sensitive to the treatment (and definition) of the soil surface microroughness, and thus, require an adequate determination of the soil surface roughness length scales for accurately modeling the hydrologic response of hillslopes. To the extent that microscale processes are considered significant, we argue that such models should adequately capture the increasing and decreasing trends in soil microroughness during all stages of a storm event in order to accurately predict local response to rainfall. The extent to which the increase in RR recorded herein can affect erosion processes is not yet known. However, it has been noted that different values of RR can affect flow pathways and runoff, which consequently can affect erosion processes (Gómez and Nearing, 2005).~~

[revised manuscript text omitted]

* The Vázquez et al. (2008) study looked at RR evolution under successive rainfall events, unlike the other two studies.  Post-rainfall RR data presented  for Vázquez et al. (2008) are those that were determined on completion of the last rainfall succession in each experiment.

**Table 2:** Summary of the rainfall induced change in the crossover length, the Markov-Gaussian variance length scale and limiting difference indices for the experimental tests of this study.

| Rainfall Intensity (mm/h) | Cumulative Rainfall (mm) | Pre-rainfall value | Post-rainfall value | Index Ratio |
|---|---|---|---|---|
| | | | *l (mm)* | |
| 30 | 150 | 0.71 | 0.73 | 1.03 |
| 60 | 300 | 0.09 | 0.20 | 2.13 |
| 75 | 375 | 0.15 | 0.39 | 2.56 |
| | | | *σ (mm)* | |
| 30 | 150 | 1.19 | 1.63 | 1.37 |
| 60 | 300 | 0.42 | 1.52 | 3.62 |
| 75 | 375 | 0.31 | 1.43 | 4.56 |
| | | | *LD (mm)* | |
| 30 | 150 | 0.79 | 0.87 | 1.10 |
| 60 | 300 | 0.26 | 0.87 | 3.39 |
| 75 | 375 | 0.15 | 0.71 | 4.84 |

|  |  |  |  |  |  |  |  |
|---|---|---|---|---|---|---|---|
|  |  |  |  |  |  |  |  |
|  |  |  |  |  |  |  |  |
|  |  |  |  |  |  |  |  |